# SODA: STREAM OUT-OF-DISTRIBUTION ADAPTATION

## ABSTRACT

In open-context environments, machine learning models require out-of-distribution (OOD) awareness to ensure safe operation. However, existing OOD detection approaches have primarily focused on the offline setting, where OOD detectors remain static and fixed after deployment. This limits their ability to perform in real-world environments with unknown and ever-shifting out-of-distribution data. To address this limitation, we propose a novel *online OOD detection* framework that allows for continuous adaptation of the OOD detector. Our framework updates the ID classifier and OOD detector sequentially, based on samples observed from the deployed environment, and minimizes the risk of incorrect OOD predictions at each timestep. Unlike traditional offline OOD detection methods, our online framework provides the adaptivity and practicality needed for real-world environments. Theoretical analysis demonstrates that our algorithm provably achieves sub-linear regret and converges to the optimal OOD detector over time. Empirical evaluation in various environments shows that our online OOD detector significantly outperforms offline methods, highlighting the superiority of our framework for real-world applications of OOD detection.

## 1 INTRODUCTION

Machine learning systems today must operate amid increasingly dynamic and open environments. One important characteristic of the deployed environment is the occurrence of out-of-distribution (OOD) samples which are not taught to the model during training time. It is generally acknowledged that machine learning models, particularly in the supervised setting, can be brittle and lack necessary awareness of OOD data (Nguyen et al., 2015). Needless to argue, the safety of future ML systems depends critically on developing reliable algorithms for detecting OOD samples in the wild.

In literature, the defacto procedure for OOD detection is to first train an offline supervised model on the in-distribution (ID) data, and then derive the OOD detector based on the learned classifier (Yang et al., 2021). Numerous algorithms have been developed over the years, focusing on either training-time regularization (for facilitating a conservative decision boundary around ID data) (Bevandić et al., 2018; Lee et al., 2018a; Malinin & Gales, 2018; Tao et al., 2023; Wang et al., 2023) or test-time scoring functions (for deriving statistical estimates to separate ID and OOD data) (Hendrycks & Gimpel, 2017; Lakshminarayanan et al., 2017; Lee et al., 2018b; Liang et al., 2018; Liu et al., 2020; Huang et al., 2021; Sehwag et al., 2021; Sun et al., 2021; 2022; Djurisic et al., 2023). However, prior works suffer from a common limitation—the OOD detector is assumed *static* and fixed in the deployed environment. This restrictive setting hinders OOD detection in real-world *dynamic* environments, where out-of-distribution is unknown and can be subject to continuous changes over time. A concrete example is that of a self-driving car deployed in the real world. The incoming OOD observations may be contingent on the location of the car at a specific time. In particular, the car may encounter shifting OOD observations such as snow mounds, dirt mounds, or desert mounds as the car drives across different terrains in the real world. This motivates the need for us to shift our perspective on OOD detection from the previous offline setting to an online setting.

**Algorithmic Contributions** In this paper, we propose and formalize a new *online OOD detection* framework, which enables OOD detectors to continuously adapt with respect to the online environment. A driving idea of our framework is to leverage data from the deployed environment, which can arrive sequentially from either ID or OOD. A learner's goal is to incrementally update the ID classifier and OOD detector based on the environment observations, and minimize the risk of making

incorrect ID and OOD predictions at each timestep. In contrast to the classic offline OOD detection setting, our online framework brings the benefits of adaptivity and practicality.

- **Adaptivity**: Online OOD detection enables the OOD detector to constantly and autonomously evolve with respect to the underlying distributional shifts of the deployed environment.
- **Practicality**: Online OOD detection efficiently manages a continuous stream of readily available OOD data, offering a pragmatic approach to real-world OOD detection and eliminating the necessity for manual dataset collection offline.

Beyond formalizing the framework, we also present a realization of an online OOD detection algorithm called Stream Out-of-Distribution Adaptation (SODA). The algorithm flow works as follows. In each round of online interaction, the model receives an environment instance and makes a prediction on whether the observed sample belongs to ID or OOD. After making the prediction, the model receives feedback about the instance from the environment. Based on the feedback, the model incurs some loss dependent on whether the sample is ID or OOD and updates its parameters. In addition, we also provide a straightforward unsupervised extension for SODA, enabling SODA to tackle the online setting without necessitating any environment feedback. Importantly, our SODA algorithm can also be adapted to operate with various OOD scoring functions through different instantiations of the ID and OOD loss functions.

**Theoretical and Empirical Insights** SODA enjoys both strong theoretical guarantees and empirical performance. Theoretically, we provably show that SODA can achieve sub-linear regret (Section 3.3). In the context of online OOD detection, regret measures the ID and OOD loss suffered by the learner over a sequence of $T$ rounds, relative to the best model in hindsight. From a theoretical perspective of regret minimization, our regret bound suggests that the online OOD detector converges to the optimal detector at a sub-linear rate and makes a decreasing number of OOD prediction mistakes as time grows. On a comprehensive collection of stationary and non-stationary environments, we empirically validate our theory that the online OOD detector is sub-linearly converging to the optimal OOD detector. Furthermore, we also empirically show that our online OOD detection algorithm significantly outperforms the offline counterparts (Section 4). In particular, we observe significant improvements in the challenging ImageNet-1k benchmark, reducing the final average FPR95 by **18.54**% when compared to the current best method WOODS (Katz-Samuels et al., 2022). These results emphasize the advantages of our online OOD detection framework, which enables continuous adaptation of the OOD detector as OOD observations emerge.

## 2 AN ONLINE FORMULATION OF OUT-OF-DISTRIBUTION DETECTION

We begin with a formal definition and overview of the *online out-of-distribution detection* problem. In the following sections, we first formally define the online environment (in Section 2.1), and then provide an overview of the online OOD detection framework (in Section 2.2).

### 2.1 ENVIRONMENT SPECIFICATION

Online OOD detection considers the setting where a multi-class classifier $f : \mathbb{R}^d \to \mathbb{R}^k$, trained on the in-distribution data $\mathbf{x} \in \mathbb{R}^d$ with $k$ number of classes, is deployed in an online environment where distributional shifts can occur. In particular, data from the environment arrives sequentially from either ID or OOD. The marginal distribution of the environment data is furthermore subject to changes over time. A learner's goal is to learn and update the OOD detector to make minimal ID or OOD classification mistakes at every step. Formally, we denote the marginal distribution of the environment data as

$$\mathcal{Q}_t^{\text{env}} = (1 - \pi_t)\mathcal{P}^{\text{in}} + \pi_t \mathcal{P}_t^{\text{out}}, \tag{1}$$

where $\mathcal{P}^{\text{in}}$ and $\mathcal{P}_t^{\text{out}}$ denote the marginal distributions of ID and OOD respectively, and $\pi_t \in [0, 1]$ is a probability denoting the ID and OOD mixture percentage at time $t$. It is important to note that none of the underlying distributions $\mathcal{P}^{\text{in}}$, $\mathcal{P}_t^{\text{out}}$ or mixture percentage $\pi_t$ is known to the learner. The learner only interacts with observations that the environment produces during the online interaction. Furthermore, our formulation generalizes prior in-the-wild OOD formulations (Katz-Samuels et al., 2022), which implicitly restricts OOD to be stationary with respect to time.

## 2.2 ONLINE OUT-OF-DISTRIBUTION DETECTION FRAMEWORK: AN OVERVIEW

Now we formally define the online OOD detection framework. We summarize the general procedure of online OOD detection in Algorithm 1. The goal of online OOD detection is to (1) incrementally update the model $f_t$ and OOD detector $g_t$ based on environmental observations and (2) minimize errors in ID classification and OOD detection at each timestep $t$. At each timestep $t$, a deployed model interacts with an online environment $\mathcal{Q}_t^{\text{env}}$, which generates sequential mixtures of ID and OOD samples. In each round, the model makes a prediction on whether the environment observation $\mathbf{x}_t \in \mathbb{R}^d$ belongs to ID or OOD, as well as the class prediction if the observation is ID. Following this, the model receives feedback about the instance from the environment, incurring an OOD loss dependent on whether $\mathbf{x}_t$ is ID or OOD. The ID loss is contingent on the class label provided by the environmental feedback. Based on the loss, we then update both the ID classifier from $f_t$ to $f_{t+1}$ and the OOD detector from $g_t$ to $g_{t+1}$. In certain real-world scenarios, obtaining reliable environmental feedback may be challenging. To overcome this challenge, we further offer an unsupervised extension that operates without any feedback from the environment (see Appendix D).

---

**Algorithm 1** Online Out-of-Distribution Detection: An Overview

---

1: Initialize the multi-class classifier as $f_1 : \mathbb{R}^d \to \mathbb{R}^k$
2: Define the OOD detector $g_1 = S \circ f_1 : \mathbb{R}^d \to \{\text{ID}, \text{OOD}\}$ where $S$ is the OOD scoring function
3: **for** $t = 1, \ldots, T$ **do**
4:     Receive environment instance $\mathbf{x}_t$
5:     Predict whether $\mathbf{x}_t$ is ID or OOD using $g_t$
6:     Environment provides feedback on $\mathbf{x}_t$
7:     **if** $\mathbf{x}_t$ is OOD **then**
8:         Suffer OOD loss on $\mathbf{x}_t$
9:     **else**
10:         Suffer ID classification loss on $\mathbf{x}_t$
11:     **end if**
12:     Update the classifier from $f_t$ to $f_{t+1}$
13:     Update the OOD detector from $g_t$ to $g_{t+1}$
14: **end for**

---

## 3 METHODOLOGY

In this section, we provide a concrete algorithmic realization of our online OOD detection framework in Section 3.1. Then, in Section 3.2, we define regret in the context of online OOD detection, followed by a theoretical analysis of the regret bound in Section 3.3.

### 3.1 SODA: STREAM OUT-OF-DISTRIBUTION ADAPTATION

We present a realization of an Online OOD Detection algorithm called Stream Out-of-Distribution Adaptation (SODA). We summarize the complete SODA algorithm in Algorithm 2. At every iteration, the algorithm takes a step from the current model to a new updated model, in the direction of the gradient of the loss function. The loss is contingent on the environmental feedback, denoted through $y_t \in \{-1, \ldots, k-1\}$, where $y_t = -1$ indicates that $\mathbf{x}_t$ is OOD. SODA can also be extended to the unsupervised setting where the environment does not provide the feedback $y_t$ (see Appendix D).

Additionally, depending on the chosen OOD scoring function, different loss functions can be used to instantiate $\mathcal{L}_t^{\text{id}}$ and $\mathcal{L}_t^{\text{ood}}$. In this paper, our primary objective is to establish an initial online OOD detection algorithm, to serve as a prospective future baseline. Therefore, for the remainder of our study, we will employ the maximum softmax probability as the default OOD scoring function, as it is a widely recognized baseline in the field of OOD detection. *It is important to highlight that* SODA *provides the flexibility to accommodate various OOD scoring functions by modifying the instantiations of $\mathcal{L}_t^{id}$ and $\mathcal{L}_t^{ood}$. We provide examples of different instantiations in Section 4.3*, including an energy-based approach.

**Out-of-Distribution Loss Function**   When the observed sample from the online environment is OOD, the loss function penalizes overconfidence when observing OOD samples. At every round

---

**Algorithm 2** Stream Out-of-Distribution Adaptation (SODA)

---

1: **Input hyperparameter:** learning rate $\eta$
2: Train the initial classifier $f_1 : \mathbb{R}^d \to \mathbb{R}^k$ on the ID task
3: Define the initial OOD detector $g_1 = S \circ f_1 : \mathbb{R}^d \to \{\text{ID}, \text{OOD}\}$, and $S$ is a scoring function
4: **for** $t = 1, \dots, T$ **do**
5:     Receive environment instance $\mathbf{x}_t$
6:     Predict whether $\mathbf{x}_t$ is ID or OOD using $g_t$.
7:     Environment reveals $y_t$
8:     **if** $y_t = -1$ ($\mathbf{x}_t$ is OOD) **then**
9:         Suffer OOD loss $\mathcal{L}_t^{\text{ood}}(\boldsymbol{\theta}_t; \mathbf{x}_t)$ defined in Equation 2
10:        Update model parameters $\boldsymbol{\theta}_{t+1} = \boldsymbol{\theta}_t - \eta \nabla_\theta \mathcal{L}_t^{\text{ood}}(\boldsymbol{\theta}_t; \mathbf{x}_t)$
11:     **else**
12:         Suffer ID classification loss $\mathcal{L}_t^{\text{id}}(\boldsymbol{\theta}_t; (\mathbf{x}_t, y_t))$ defined in Equation 3
13:        Update model parameters $\boldsymbol{\theta}_{t+1} = \boldsymbol{\theta}_t - \eta \nabla_\theta \mathcal{L}_t^{\text{id}}(\boldsymbol{\theta}_t; (\mathbf{x}_t, y_t))$
14:     **end if**
15:     Update OOD detector $g_{t+1} := \mathbb{I}(\max_i \frac{\exp(f_{t+1}^{(i)}(\mathbf{x}; \boldsymbol{\theta}_{t+1}))}{\sum_{j=1}^k \exp(f_{t+1}^{(j)}(\mathbf{x}; \boldsymbol{\theta}_{t+1}))} > \lambda)$.
16:                              ▷ 1 represents the positive class (ID), and 0 indicates OOD.
17: **end for**
18: **Return** $f_T$ and $g_T$

---

$t$, given a neural network $f_t$ parameterized by $\boldsymbol{\theta}_t$, the network takes an input $\mathbf{x}_t \in \mathbb{R}^d$ and maps it to a set of logits $f_t(\mathbf{x}_t; \boldsymbol{\theta}_t) \in \mathbb{R}^k$, where $k$ is the number of distinct ID classes. With this setup, the per-sample OOD loss $\mathcal{L}_t^{\text{ood}}$ can be defined through the cross-entropy between the prediction and target uniform vector

$$\mathcal{L}_t^{\text{ood}}(\boldsymbol{\theta}_t; \mathbf{x}_t) = -\beta \sum_{i=1}^k \frac{1}{k} \log \left( \frac{\exp(f_t^{(i)}(\mathbf{x}_t; \boldsymbol{\theta}_t))}{\sum_{j=1}^k \exp(f_t^{(j)}(\mathbf{x}_t; \boldsymbol{\theta}_t))} \right), \tag{2}$$

where $f_t^{(i)}(\mathbf{x}_t; \boldsymbol{\theta}_t)$ denotes the $i$-th element of $f_t(\mathbf{x}_t; \boldsymbol{\theta}_t)$, and $\beta$ is a coefficient that modulates the weight of the penalty. We provide additional analysis on $\beta \in [0, 1]$ in Appendix E.

**In-Distribution Loss Function**    Given that the observed sample $\mathbf{x}_t$ from the online environment is ID, the per-sample classification loss $\mathcal{L}_t^{\text{id}}$ is defined using the cross-entropy (CE) loss

$$\mathcal{L}_t^{\text{id}}(\boldsymbol{\theta}_t; (\mathbf{x}_t, y_t)) = -\log \left( \frac{\exp(f_t^{(y_t)}(\mathbf{x}_t; \boldsymbol{\theta}_t))}{\sum_{i=1}^k \exp(f_t^{(i)}(\mathbf{x}_t; \boldsymbol{\theta}_t))} \right), \tag{3}$$

where $y_t$ denotes the ID classification label revealed by the environment. Next, we provide theoretical guarantees for SODA, by formally defining and bounding the regret in Section 3.2 and Section 3.3.

## 3.2 Defining Regret

In the context of online OOD detection, regret is a performance metric that measures the cumulative loss suffered by the learner over a sequence of $T$ rounds, relative to the best model in hindsight. Formally, we define regret as

$$\text{regret} = \sum_{t=1}^T \mathcal{L}_t(\boldsymbol{\theta}_t) - \min_{\boldsymbol{\theta} \in \Theta} \sum_{t=1}^T \mathcal{L}_t(\boldsymbol{\theta}), \tag{4}$$

where $\mathcal{L}_t$ is defined in Section 3.1 as either $\mathcal{L}_t^{\text{ood}}$ or $\mathcal{L}_t^{\text{id}}$ dependent on time $t$, and $\min_{\boldsymbol{\theta} \in \Theta} \sum_{t=1}^T \mathcal{L}_t(\boldsymbol{\theta})$ can be interpreted as an optimal model that has access to the true distribution of the incoming samples. Different from classic online learning (Hoi et al., 2021), our regret definition captures the performance of both OOD detection and ID classification, due to the fact that the loss $\mathcal{L}_t$ can be instantiated by both ID and OOD samples. In the context of regret minimization, if an online algorithm attains sub-linear regret ($o(T)$), it signifies that $\lim_{T \to \infty} \text{regret}/T = 0$. This convergence implies that the learner's average performance gradually approaches that of the best possible learner. Next, we formally provide the regret bound, showing that our algorithm can indeed achieve sub-linear regret.

## 3.3 REGRET BOUND

**Theorem 3.1 (Sub-linear Regret Bound)** *Under conditions that are commonly found in online convex optimization (c.f. Appendix B.1), the regret for our online OOD detection algorithm can be upper bounded by*

$$regret = \sum_{t=1}^{T} \mathcal{L}_t(\boldsymbol{\theta}_t) - \min_{\boldsymbol{\theta} \in \Theta} \sum_{t=1}^{T} \mathcal{L}_t(\boldsymbol{\theta}) \tag{5}$$

$$\leq \sqrt{TB} \cdot \left( (\sup_{\mathbf{x} \sim \mathcal{P}^{\text{in}}} \|\mathbf{x}\|_2) \sqrt{\frac{(1 - \tilde{\pi})(k-1)}{k}} + (\sup_{\mathbf{x} \sim \mathcal{P}^{\text{out}}} \|\mathbf{x}\|_2) \sqrt{\frac{\tilde{\pi}\beta^2(k-1)}{k}} \right) \tag{6}$$

$$\approx o(T), \tag{7}$$

*where $\tilde{\pi}$ is an empirical estimate of the mixture ratio, $\beta$ is a constant hyperparameter, $k$ is the number of ID classes, and $T$ is the total timestep of the online interaction. Additional proof details can be found in Appendix B.1.*

**Theoretical Insights**   Firstly, we note that SODA is sub-linear in regret with no assumption on the stationarity of the environment OOD. This means that the learner is converging to the optimal function at a sub-linear rate. In other words, we can guarantee that our model is learning to be less overconfident on the OOD samples. This also implies that our OOD detector makes a decreasing number of mistakes as time $t$ grows.

Observing Equation 6, we can also dissect the regret bound into two individual parts, with Term 1 capturing the difficulty of learning from the ID samples and Term 2 capturing the difficulty of learning from the OOD samples.

$$\text{regret} \leq \sqrt{TB} \cdot \left( \underbrace{(\sup_{\mathbf{x} \sim \mathcal{P}^{\text{in}}} \|\mathbf{x}\|_2) \sqrt{\frac{(1 - \tilde{\pi})(k-1)}{k}}}_{\substack{\text{In-distribution} \\ \text{Loss Bound} \\ \textbf{(Term 1)}}} + \underbrace{(\sup_{\mathbf{x} \sim \mathcal{P}^{\text{out}}} \|\mathbf{x}\|_2) \sqrt{\frac{\tilde{\pi}\beta^2(k-1)}{k}}}_{\substack{\text{Out-of-distribution} \\ \text{Loss Bound} \\ \textbf{(Term 2)}}} \right) \tag{8}$$

In particular, Term 1 and Term 2 are intricately linked to the underlying mixture ratio and the associated losses derived from the observed ID and OOD samples. In practice, given a sufficiently well-trained model, the influence of SODA is primarily determined by our second term. Namely, the likelihood of the environment generating an OOD sample $\pi_t$ intertwined with the complexity of learning the environment OOD. Our theory has practical implications too. For example, our guarantee ensures the optimality of SODA under linear probing, which is a widely employed technique in online learning, continual learning, and transfer learning. Thus, our theoretical findings have direct parallels in various real-world applications of online OOD detection.

## 4 EXPERIMENTAL VALIDATIONS

In this section, we conduct a comprehensive set of evaluations. First, we present dataset and environment details in Section 4.1. Then we showcase our main empirical results in Section 4.2. Finally, we present ablation studies in Section 4.3. Additional details on the experimental setup and evaluation metrics can be found in Appendix C. We present results for unsupervised extension of SODA in Appendix D, and provide further supplementary experiments in Appendix E.

### 4.1 EXPERIMENTAL SETUP

**Datasets**   We consider both small-scale and large-scale ID datasets: CIFAR-10 (Krizhevsky et al., 2009) and ImageNet-1k (Deng et al., 2009). For OOD datasets, we leverage SVHN (Netzer et al., 2011b), LSUN-R (Yu et al., 2015), Places365 (Zhou et al., 2017), and Textures (Cimpoi et al., 2014) with respect to CIFAR-10 as the ID dataset. For ImageNet-1k, we use the large-scale OOD detection benchmark (Huang & Li, 2021), and evaluate against iNaturalist (Van Horn et al., 2018), SUN (Xiao et al., 2010), Places (Zhou et al., 2017), and Textures (Cimpoi et al., 2014). The CIFAR-10 training dataset is split into two sets: 10,000 images used for training the initial ID classifier (to be deployed in an online environment), and 40,000 images used to simulate

| $P_{in}$ (Model) | Method | SVHN | | LSUN-R | | Textures | | Places365 | | Average | | Average ID Acc. |
|---|---|---|---|---|---|---|---|---|---|---|---|---|
| | | FPR↓ | AUROC↑ | FPR↓ | AUROC↑ | FPR↓ | AUROC↑ | FPR↓ | AUROC↑ | FPR↓ | AUROC↑ | |
| | | **Offline OOD Detection (using ID data only)** | | | | | | | | | | |
| CIFAR-10 (ResNet-34) | MSP | 50.59 | 92.35 | 40.35 | 94.10 | 55.73 | 89.76 | 59.27 | 88.16 | 51.49 | 91.09 | 93.13 |
| | ODIN | 36.63 | 92.74 | 11.74 | 97.55 | 45.55 | 87.29 | 41.58 | 88.97 | 33.88 | 91.64 | 93.13 |
| | Energy | 24.52 | 95.35 | 14.31 | 97.22 | 47.66 | 87.24 | 41.61 | 89.62 | 32.03 | 92.36 | 93.13 |
| | Mahalanobis | 18.09 | 94.83 | 2.59 | 99.48 | 45.46 | 88.09 | 71.77 | 80.04 | 34.48 | 90.61 | 93.13 |
| | ReAct | 20.87 | 95.64 | 10.48 | 97.76 | 41.52 | 90.24 | 41.46 | 90.20 | 28.58 | 93.83 | 93.13 |
| | KNN | 33.05 | 94.38 | 29.70 | 95.20 | 30.85 | 94.81 | 45.30 | 90.91 | 35.28 | 93.64 | 93.13 |
| | ASH | 19.05 | 96.61 | 9.61 | 97.83 | 32.84 | 92.10 | 43.56 | 90.65 | 26.27 | 94.30 | 93.13 |
| | SSD+ | 6.79 | 98.34 | 12.60 | 97.27 | 16.48 | 97.09 | 33.96 | 93.12 | 17.46 | 96.46 | 93.91 |
| | KNN+ | 5.07 | 98.72 | 6.25 | 98.46 | 9.53 | 97.74 | 25.34 | 95.09 | 11.55 | 97.50 | 93.91 |
| | | **Offline OOD Detection (using ID and pre-collected environment data)** | | | | | | | | | | |
| | WOODS | 0.22 | 99.95 | 0.18 | 99.94 | 11.92 | 97.82 | 24.02 | 95.01 | 9.09 | 98.18 | 92.81 |
| | | **Online OOD Detection** | | | | | | | | | | |
| | SODA (Avg) | **2.29**±0.2 | **99.65**±0.0 | **1.73**±0.1 | **99.74**±0.0 | **4.97**±0.1 | **99.06**±0.0 | **8.54**±0.9 | **97.89**±0.7 | **4.38**±0.3 | **99.09**±0.2 | 93.49±0.1 |
| | SODA (Final) | **0.15**±0.3 | **99.97**±0.1 | **0.09**±0.1 | **99.97**±0.0 | **2.09**±0.2 | **99.63**±0.1 | **5.33**±1.1 | **98.25**±0.7 | **1.92**±0.4 | **99.46**±0.2 | 93.55±0.3 |
| | | iNaturalist | | SUN | | Textures | | Places | | Average | | |
| | | **Offline OOD Detection (using ID data only)** | | | | | | | | | | |
| ImageNet-1k (ResNet-101) | MSP | 58.30 | 85.90 | 68.78 | 81.13 | 64.29 | 80.41 | 71.52 | 80.07 | 62.72 | 81.88 | 77.52 |
| | ODIN | 53.18 | 88.80 | 55.47 | 86.94 | 48.87 | 87.22 | 62.33 | 84.59 | 54.96 | 86.89 | 77.52 |
| | Energy | 59.09 | 87.65 | 54.36 | 87.34 | 51.42 | 86.39 | 61.49 | 85.00 | 56.59 | 86.60 | 77.52 |
| | Mahalanobis | 88.73 | 59.34 | 91.46 | 54.89 | 95.03 | 40.65 | 76.97 | 73.78 | 88.05 | 57.17 | 77.52 |
| | ReAct | 23.50 | 95.16 | 21.83 | 95.57 | 44.23 | 89.97 | 39.15 | 90.26 | 32.18 | 92.74 | 77.52 |
| | KNN | 64.29 | 84.74 | 70.92 | 80.83 | 31.30 | 89.25 | 73.78 | 79.94 | 60.07 | 83.69 | 77.52 |
| | ASH | 21.85 | 95.76 | 28.93 | 93.48 | 25.83 | 94.60 | 42.03 | 89.86 | 29.66 | 93.43 | 77.52 |
| | SSD+ | 54.77 | 88.73 | 75.65 | 78.23 | 39.84 | 87.61 | 76.22 | 75.06 | 61.62 | 82.41 | 80.20 |
| | KNN+ | 27.85 | 95.03 | 43.76 | 89.78 | 20.79 | 94.88 | 55.54 | 84.94 | 36.99 | 91.16 | 80.20 |
| | | **Offline OOD Detection (using ID and pre-collected environment data)** | | | | | | | | | | |
| | WOODS | 17.20 | 96.95 | 10.32 | 97.51 | 19.81 | 95.23 | 34.92 | 92.19 | 20.56 | 95.47 | 76.78 |
| | | **Online OOD Detection** | | | | | | | | | | |
| | SODA (Avg) | **2.99**±0.1 | **99.27**±0.0 | **5.93**±0.2 | **98.43**±0.0 | **7.76**±0.2 | **97.93**±0.0 | **9.15**±0.3 | **97.60**±0.1 | **6.46**±0.2 | **98.31**±0.0 | 77.85±0.2 |
| | SODA (Final) | **0.71**±0.2 | **99.83**±0.0 | **2.19**±0.3 | **99.45**±0.1 | **2.07**±0.2 | **99.50**±0.0 | **3.12**±0.3 | **99.25**±0.1 | **2.02**±0.3 | **99.51**±0.1 | 77.96±0.5 |

Table 1: **Main results.** Comparison of OOD detection performance between offline and online OOD detection methods for CIFAR-10 and ImageNet-1k. The results of our online method are averaged over 5 random runs. We report both the average and final performance as specified in Appendix C.4. Superior results are in **bold**.

the online environment mixture (see below for details). Similarly, the ImageNet-1k dataset is split into two sets with 1,241,167 images used for ID pre-training and 40,000 images used to simulate the online environment mixture.

**Online Environments** To simulate the online environment, we consider both *stationary* and *non-stationary* environments, which contain a mixture of ID and OOD samples. More specifically, stationary online environment can be modeled as $\mathcal{Q}_t^{\text{env}} = (1 - \pi)\mathcal{P}^{\text{in}} + \pi\mathcal{P}^{\text{out}}$, where $\mathcal{P}^{\text{out}}$ remains unchanged over the course of the online interaction. To simulate this, we mix a subset of ID data (as $\mathcal{P}^{\text{in}}$) with the OOD dataset (as $\mathcal{P}^{\text{out}}$) under the default $\pi = 0.2$, which reflects the practical scenario where most data would remain ID. Take SVHN as an example, we deploy the model with CIFAR+SVHN as the online environment and test on SVHN as OOD. We simulate this for all OOD datasets and provide analysis of differing $\pi \in \{0.0, 0.1, \ldots, 1.0\}$ in Section 4.3.

To simulate a non-stationary environment, we leverage all OOD datasets {SVHN, LSUN Resized, Textures, Places365} as possible $\mathcal{P}_t^{\text{out}}$ distributions. At every $\Delta = 4{,}800$ timestamp, we switch the OOD dataset used in the online mixture data and set a new mixture ratio by picking uniformly randomly $\pi_t \in [0, 1]$. For example, if the current OOD dataset is SVHN with $\pi_t \sim \mathcal{U}[0, 1]$, we switch the OOD dataset to LSUN at timestep $\Delta + t$ with $\pi_{\Delta+t} \sim \mathcal{U}[0, 1]$. We also provide analysis of differing $\Delta \in \{800, 1600, \ldots, 5600\}$ in Appendix E. Additionally, for both the stationary and non-stationary environments, the online learner conducts $T = 30{,}000$ rounds of interaction.

## 4.2 RESULTS

**Comparison with Offline Methods** In Table 1, we draw direct comparisons between traditional offline OOD detection methods (which remains fixed with respect to the pre-trained ID classifier) and our online OOD detection method SODA (which incrementally adapts to the observed online sample). In particular, we compare with a wide range of offline OOD detection methods, including MSP (Hendrycks & Gimpel, 2017), ODIN (Liang et al., 2018), Energy (Liu et al., 2020), Mahalanobis (Lee et al., 2018b), ReAct (Sun et al., 2021), KNN (Sun et al., 2022), SSD+ (Sehwag et al., 2021), KNN+ (Sun et al., 2022), and ASH (Djurisic et al., 2023). For a fair comparison, both offline and online models are pre-trained on the same dataset as described in Section 4.1. Results for both

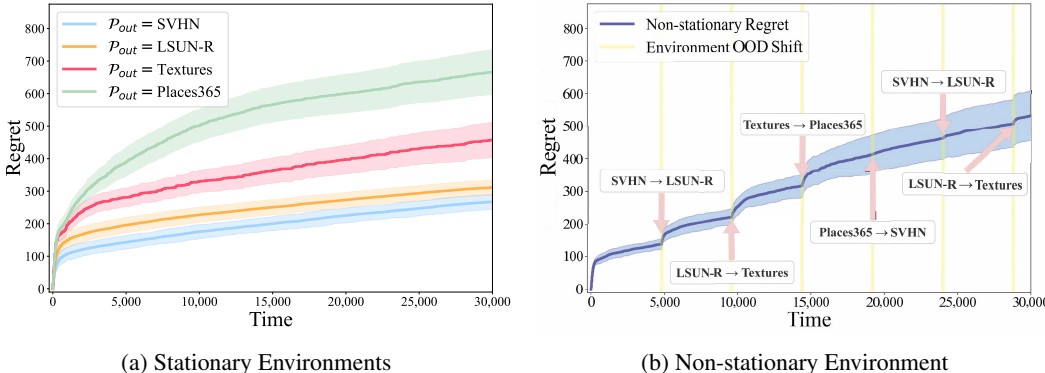

Figure 1: **Regret** of SODA under (a) stationary environments and (b) non-stationary environments. All results are averaged over 5 random runs. See Section 4.1 for online environment setup details.

SSD+ and KNN+ are based on a model trained with the SupCon (Khosla et al., 2020) loss, using the same ID pre-training data split as other baselines. Additional results with ReAct and ASH are obtained with Energy as the OOD detection score. We can see that our online algorithm significantly outperforms the offline counterparts. Compared to the best offline method KNN+, SODA reduces the final average FPR95 by **9.63**% in the CIFAR-10 ID setting. This further emphasizes the advantage of our online OOD detection framework, which enables continuous adaptation of the OOD detector. We also perform evaluations on unseen OOD test datasets, which differ from the environment OOD $\mathcal{P}^{\text{out}}$. The results can be found in Appendix E, where we also see strong performance.

**Comparison with Method Using Offline Environment Data**  In Table 1, we additionally present a comparison between SODA and WOODS (Katz-Samuels et al., 2022), a state-of-the-art offline method that leverages environment data. We note a few key differences: (1) WOODS trains an OOD-aware classifier using a pre-collected *offline* wild mixture dataset and a labeled ID dataset. In other words, WOODS is trained offline using all the environment observations accumulated over the $T = 30,000$ interaction. This is not directly comparable with our online OOD detection setting, as we do not assume the model has access to the environment samples in advance. Simply put, our setting enables learning on the fly, while WOODS does not. (2) WOODS also requires a copy of the pre-training ID dataset, which is not required in our online OOD detection setting. Therefore, performance comparisons between these two methods should be interpreted with these differences in mind. From Table 1, we can see that our method reduces the final average FPR95 by **7.17**% in the CIFAR-10 ID setting and **18.54**% in the ImageNet-1k ID setting. This again demonstrates the advantages of our online OOD detection method.

**Sub-linear Regret for Stationary Environments**  We provide a regret plot in Figure 1a, validating that our online algorithm is indeed sub-linearly converging to the optimal detector. This is encouraging since the online detector's performance will be close to that of the best possible model as time increases. For a comprehensive evaluation, we consider four different stationary environments, which are simulated by mixing CIFAR-10 with one of the OOD datasets: SVHN, LSUN-R, Textures, Places365. The details of simulating the environments are specified in Section 4.1. Taking SVHN ($\mathcal{P}^{\text{out}}$) as an example, our online learner converges reasonably fast, with approximately 800 timesteps (i.e., 800 samples). Places365 is the most challenging OOD dataset, which converges more slowly than SVHN, but nonetheless, displays sub-linearity in regret.

**Results for Non-Stationary Environments**  Going beyond stationary environments, we further show that SODA can handle more challenging scenarios in non-stationary environments. To the best of our knowledge, our work is the first to demonstrate such capability in adapting to ever-changing OOD distributions. We present results in Figure 1b, where we show the regret under the non-stationary online environment as specified in Section 2.1. Each vertical line (colored in yellow) marks the timestamp with an OOD distribution change. We would like to highlight three observations: (1) Within each time window $\Delta$ (i.e., any two consecutive vertical lines), we note that the online OOD detector can converge sub-linearly in regret. This is consistent with our previous observations when the environment distribution is relatively stationary. (2) At each changing point of $\mathcal{P}^{\text{out}}$ marked by the vertical lines, the regret can have a sudden spike (e.g., at timestep 4,800)

| $P_{in}$ (Model) | Instantiation | Evaluated OOD Datasets | | | | | | | | | | Average ID Acc. |
|---|---|---|---|---|---|---|---|---|---|---|---|---|
| | | SVHN | | LSUN-R | | Textures | | Places365 | | Average | | |
| | | FPR ↓ | AUROC ↑ | FPR ↓ | AUROC ↑ | FPR ↓ | AUROC ↑ | FPR ↓ | AUROC ↑ | FPR ↓ | AUROC ↑ | |
| | | | | | | Online OOD Detection | | | | | | |
| CIFAR-10 (ResNet-34) | SODA+MSP (Avg) | $2.29^{\pm0.2}$ | $99.65^{\pm0.0}$ | $1.73^{\pm0.1}$ | $99.74^{\pm0.0}$ | $4.97^{\pm0.1}$ | $99.06^{\pm0.0}$ | $8.54^{\pm0.9}$ | $97.89^{\pm0.7}$ | $4.38^{\pm0.3}$ | $99.09^{\pm0.2}$ | $93.49^{\pm0.1}$ |
| | SODA+MSP (Final) | $0.15^{\pm0.3}$ | $99.97^{\pm0.1}$ | $0.09^{\pm0.1}$ | $99.97^{\pm0.0}$ | $2.09^{\pm0.2}$ | $99.63^{\pm0.1}$ | $5.33^{\pm1.1}$ | $98.25^{\pm0.7}$ | $1.92^{\pm0.4}$ | $99.46^{\pm0.2}$ | $93.55^{\pm0.3}$ |
| | SODA+ODIN (Avg) | $1.94^{\pm0.1}$ | $99.61^{\pm0.0}$ | $1.26^{\pm0.1}$ | $99.84^{\pm0.0}$ | $4.84^{\pm0.1}$ | $98.91^{\pm0.0}$ | $8.67^{\pm1.0}$ | $97.94^{\pm0.7}$ | $4.18^{\pm0.4}$ | $99.08^{\pm0.2}$ | $93.47^{\pm0.1}$ |
| | SODA+ODIN (Final) | $0.10^{\pm0.1}$ | $99.98^{\pm0.0}$ | $0.13^{\pm0.1}$ | $99.97^{\pm0.0}$ | $1.86^{\pm0.2}$ | $99.67^{\pm0.1}$ | $5.26^{\pm1.1}$ | $98.33^{\pm0.8}$ | $1.84^{\pm0.4}$ | $99.49^{\pm0.2}$ | $93.56^{\pm0.2}$ |
| | SODA+Energy (Avg) | $1.39^{\pm0.1}$ | $99.72^{\pm0.0}$ | $0.94^{\pm0.0}$ | $99.85^{\pm0.0}$ | $4.60^{\pm0.2}$ | $98.95^{\pm0.1}$ | $9.28^{\pm1.1}$ | $97.56^{\pm0.8}$ | $4.05^{\pm0.4}$ | $99.02^{\pm0.2}$ | $93.16^{\pm0.3}$ |
| | SODA+Energy (Final) | $0.05^{\pm0.2}$ | $99.98^{\pm0.0}$ | $0.10^{\pm0.1}$ | $99.97^{\pm0.0}$ | $1.46^{\pm0.2}$ | $99.71^{\pm0.1}$ | $5.96^{\pm1.2}$ | $98.13^{\pm0.8}$ | $1.89^{\pm0.4}$ | $99.45^{\pm0.2}$ | $93.37^{\pm0.4}$ |
| | | iNaturalist | | SUN | | Textures | | Places | | Average | | |
| | | | | | | Online OOD Detection | | | | | | |
| ImageNet-1k (ResNet-101) | SODA+MSP (Avg) | $2.99^{\pm0.1}$ | $99.27^{\pm0.0}$ | $5.93^{\pm0.2}$ | $98.43^{\pm0.0}$ | $7.76^{\pm0.2}$ | $97.93^{\pm0.0}$ | $9.15^{\pm0.1}$ | $97.60^{\pm0.1}$ | $6.46^{\pm0.2}$ | $98.31^{\pm0.0}$ | $77.85^{\pm0.2}$ |
| | SODA+MSP (Final) | $0.71^{\pm0.2}$ | $99.83^{\pm0.0}$ | $2.19^{\pm0.3}$ | $99.45^{\pm0.1}$ | $2.07^{\pm0.2}$ | $99.50^{\pm0.0}$ | $3.12^{\pm0.1}$ | $99.25^{\pm0.1}$ | $2.02^{\pm0.3}$ | $99.51^{\pm0.1}$ | $77.96^{\pm0.5}$ |
| | SODA+ODIN (Avg) | $3.60^{\pm0.2}$ | $98.73^{\pm0.1}$ | $6.16^{\pm0.2}$ | $97.73^{\pm0.1}$ | $5.28^{\pm0.3}$ | $97.93^{\pm0.1}$ | $8.45^{\pm0.3}$ | $97.51^{\pm0.1}$ | $5.87^{\pm0.3}$ | $97.98^{\pm0.1}$ | $77.80^{\pm0.2}$ |
| | SODA+ODIN (Final) | $0.63^{\pm0.3}$ | $99.82^{\pm0.1}$ | $1.15^{\pm0.3}$ | $99.54^{\pm0.1}$ | $1.84^{\pm0.3}$ | $99.68^{\pm0.1}$ | $2.18^{\pm0.4}$ | $99.55^{\pm0.2}$ | $1.45^{\pm0.3}$ | $99.65^{\pm0.1}$ | $77.89^{\pm0.4}$ |
| | SODA+Energy (Avg) | $3.26^{\pm0.1}$ | $98.94^{\pm0.0}$ | $6.67^{\pm0.3}$ | $97.58^{\pm0.1}$ | $5.57^{\pm0.2}$ | $97.95^{\pm0.1}$ | $8.66^{\pm0.1}$ | $97.52^{\pm0.1}$ | $6.04^{\pm0.2}$ | $98.00^{\pm0.1}$ | $77.10^{\pm0.3}$ |
| | SODA+Energy (Final) | $0.35^{\pm0.2}$ | $99.86^{\pm0.1}$ | $1.68^{\pm0.3}$ | $99.36^{\pm0.1}$ | $0.98^{\pm0.3}$ | $99.76^{\pm0.1}$ | $1.85^{\pm0.3}$ | $99.60^{\pm0.2}$ | $1.22^{\pm0.3}$ | $99.65^{\pm0.1}$ | $77.35^{\pm0.5}$ |

Table 2: **Results under different OOD scoring functions.** Comparison of OOD detection performance between different instantiations of the SODA algorithm under CIFAR-10 and ImageNet-1k. Results are averaged over 5 random runs and we report both the average and final performance as specified in Appendix C.4.

before sub-linearly converging. This sudden increase in regret is well corroborated by the shift in environment distribution, which necessitates the OOD detector to adapt to the new OOD distribution. (3) The overall regret is trending sub-linearly, with the spike caused by an OOD environment shift becoming less prominent as time progresses. This demonstrates the advantage of our online OOD detector, which is well-suited for real-world non-stationary environments.

## 4.3 ABLATION STUDIES

**Large-scale Evaluation** We also provide evaluations of SODA under the higher resolution, and more challenging, ImageNet-1k benchmark. We present our findings for ImageNet in Table 1, where we compare SODA with competitive OOD detection methods. We can see that the online OOD detection method still significantly outperforms offline methods. In particular, compared to the current best offline method, our method reduces the final average FPR95 by **27.64%**. This large performance gain, in the higher resolution and more realistic ImageNet setting, further supports the advantages of SODA in terms of its scalability to real-world applications.

**Instantiations of Different Scoring Functions** Central to the practical advantages of SODA is its flexibility to accommodate various OOD detection techniques. This inherent flexibility enables the integration of various OOD detection techniques into the framework of SODA, thereby elevating these traditionally static methods to operate effectively within the dynamic setting of online OOD detection. Instantiating SODA with different OOD detection techniques requires appropriate adjustments to $\mathcal{L}_t^{\text{id}}$ and $\mathcal{L}_t^{\text{ood}}$. In this ablation, we provide two illustrative examples, ODIN (Liang et al., 2018) and Energy (Liu et al., 2020), that highlight the different possible instantiations of SODA.

To instantiate SODA with ODIN, we leverage the same $\mathcal{L}_t^{\text{id}}$ and $\mathcal{L}_t^{\text{ood}}$ as defined in Section 3.1 which penalizes unwarranted confidence for OOD inputs. We choose to instantiate ODIN with these losses due to the functional consistencies between ODIN and MSP, as after temperature scaling and input perturbations ODIN still leverages low model confidence to detect OOD inputs. In contrast, to instantiate SODA with the Energy scoring function, a reformulated $\mathcal{L}_t^{\text{id}}$ and $\mathcal{L}_t^{\text{ood}}$ is necessary:

$$\mathcal{L}_t^{\text{id}}(\boldsymbol{\theta}_t; (\mathbf{x}_t, y_t)) = -\log\left(\frac{\exp(f_t^{(y_t)}(\mathbf{x}_t; \boldsymbol{\theta}_t))}{\sum_{i=1}^{k}\exp(f_t^{(i)}(\mathbf{x}_t; \boldsymbol{\theta}_t))}\right) + \beta \max(0, E(\mathbf{x}_t; \boldsymbol{\theta}_t) - m_{\text{in}})^2, \quad (9)$$

$$\mathcal{L}_t^{\text{ood}}(\boldsymbol{\theta}_t; \mathbf{x}_t) = \beta \max(0, m_{\text{out}} - E(\mathbf{x}_t; \boldsymbol{\theta}_t))^2, \quad (10)$$

where the free energy function $E(\mathbf{x}_t; \boldsymbol{\theta}_t) = -T \cdot \log\left(\sum_{i=1}^{k}\exp(f_t^{(i)}(\mathbf{x}_t; \boldsymbol{\theta}_t)/T)\right)$. In Table 2, we present an empirical comparison between the MSP, ODIN, and Energy instantiations of SODA. These empirical analyses provide insights into how SODA is able to excel across various instantiations. The consistent successes of SODA highlight its practical versatility to be utilized across a variety of different OOD detectors.

## 5 RELATED WORK

**Out-of-Distribution Detection**    A vast majority of OOD detection methods operate in the supervised learning setting, where the goal is to derive a binary classifier for distinguishing ID and OOD samples alongside an ID classification model. Attempts at addressing the problem have ranged from generative-based methods (Nalisnick et al., 2019; Ren et al., 2019; Kirichenko et al., 2020; Xiao et al., 2020; Cai & Li, 2023) to discriminative-based methods (Lakshminarayanan et al., 2017; Hsu et al., 2020; Sun et al., 2021; Fang et al., 2022; Sun & Li, 2022; Wang et al., 2022; Djurisic et al., 2023; Wu et al., 2023). In general, discriminative-based methods for OOD detection have typically been more empirically competitive, with methods being broadly grouped into output-based approaches (Hendrycks & Gimpel, 2017; Liang et al., 2018; Liu et al., 2020; Wang et al., 2021), gradient-based approaches (Huang et al., 2021), distance-based approaches (Lee et al., 2018b; Sehwag et al., 2021; Du et al., 2022a; Sun et al., 2022; Ming et al., 2023), and Bayesian approaches (Gal & Ghahramani, 2016; Lakshminarayanan et al., 2017; Maddox et al., 2019; Kristiadi et al., 2020).

Another line of OOD detection research seeks to imbue the model with OOD awareness through training time regularization on a large auxiliary OOD dataset (Bevandić et al., 2018; Malinin & Gales, 2018; Tao et al., 2023). During supervised training, the model is batch-wise trained to encourage lower confidence (Hendrycks et al., 2018; Lee et al., 2018a; Wang et al., 2023) or higher energy (Liu et al., 2020; Du et al., 2022b; Ming et al., 2022) with respect to the auxiliary OOD data. In particular, Katz-Samuels et al. attempts to train an OOD-aware classifier using a labeled ID and pre-collected *offline* wild mixture dataset. Our formulation differs in that we remove the implicit assumption that the online mixture is stationary, and we incrementally learn on top of the deployed model. Moreover, all of these methods fall into the offline learning regime, where the OOD detector remains fixed after deployment. Our framework instead focuses on the problem of how to dynamically adapt OOD detectors after deployment, specifically looking at how to learn with respect to the online environment. Our work also differs significantly from concurrent work (Yang et al., 2023), by providing (1) a rigorous formulation of the online OOD detection problem, with theoretical guarantees on regret; (2) a flexible framework that can be instantiated by different OOD detection methods and scoring functions, allowing future research to build on the framework; and (3) comprehensive experiments in both stationary and non-stationary environments that directly justify our sub-linear regret bound.

**Online Learning**    is a family of machine learning techniques that leverage a sequence of data instances, one by one at each timestep, to incrementally learn (Hazan et al., 2016; Hoi et al., 2021). Generally, online learning can be broadly categorized into online learning with full feedback (Freund & Schapire, 1997; Crammer et al., 2006; Hazan et al., 2007; Duchi et al., 2011; Dekel et al., 2012; Zhao et al., 2012; Sahoo et al., 2017; Nagabandi et al., 2018), online learning with partial feedback (Kveton et al., 2010; Goldberg et al., 2011; Loo & Marsono, 2015; Liu et al., 2018; Lykouris et al., 2018; Shen et al., 2020; Kang et al., 2023; Liu et al., 2023), and online learning with no feedback (Furao et al., 2007; Liang & Klein, 2009; Rao et al., 2019; Zhan et al., 2020; Liu & Wang, 2021). We formulate our proposed online OOD detection framework through broadly the lens of online convex optimization (Zinkevich, 2003; Shalev-Shwartz et al., 2012; Hazan et al., 2016). Furthermore, we note that a common open-ended question in online learning is handling non-stationary ID tasks (Monteleoni & Jaakkola, 2003; Nagabandi et al., 2018; Jerfel et al., 2019). However, unlike traditional online learning, our framework differs in that we deal with non-stationary OOD data in the context of OOD detection. Thus, our problem can be considered a paradigm shift from supervised OOD detection into an online OOD learning problem. Under this new formalization, we contribute methodologies, insights, and empirical validation to support the advantages of the online framework.

## 6 CONCLUSION

In this paper, we present a new framework for adaptive OOD detection, enabling the paradigm shift from a classic offline to an online setting. The capability of online OOD detection is crucial for real-world dynamic environments with constantly evolving distributions, but this online setting remained mostly unexplored prior to our work. The proposed framework leverages the sequence of data arising from the deployed environment and continuously minimizes the risk of incorrect OOD predictions. We propose a new algorithm SODA, which enjoys both strong theoretical guarantees with sub-linear regret and empirical performance that favorably outperforms offline methods. We believe the proposed framework provides a timely investigation to demonstrate the promise of online OOD learning and opens up a new door to more future works along this exciting direction.

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

## A    BROADER IMPACTS

Applications that employ ML techniques are ubiquitous, and taken together, are becoming increasingly critical in many aspects of human life. Against this backdrop, OOD detection is a rapidly growing research subarea that has the potential to fundamentally improve the deployed ML systems, hence significantly impacting society. We believe that our proposed framework represents a significant advancement in OOD detection, and gives researchers a new avenue for addressing the problems of safety and reliability for machine learning models in dynamic online environments. Given the practicality, our research is expected to have a notable impact on a broad array of applications including safe autonomous driving systems, healthcare, and scientific discovery.

## B    THEORETICAL DETAILS

**Theorem B.1 (Lipschitz Continuity)** *For analysis, consider the following assumptions which are commonly found in online convex optimization (Hazan et al., 2016; Hoi et al., 2021)*

- *$f_t$ is a single layer neural network $f_t(\mathbf{x}_t; \boldsymbol{\theta}_t) = \boldsymbol{\theta}_t^\top \mathbf{x}_t$, parameterized by $\boldsymbol{\theta}_t \in \boldsymbol{\Theta}$ where $\boldsymbol{\Theta}$ is the weight space and $\boldsymbol{\theta}_t$ is of dimensions $d \times k$, $\forall t \in \{0, 1, \ldots, T\}$.*

- *The weight space $\boldsymbol{\Theta}$ is bounded such that $\exists B \in \mathbb{R}$ where $\forall \boldsymbol{\theta}_1, \boldsymbol{\theta}_2 \in \boldsymbol{\Theta}$, $\|\boldsymbol{\theta}_1 - \boldsymbol{\theta}_2\|_F^2 \leq B$.*

- *The weight space $\boldsymbol{\Theta}$ is closed where given any sequence $\{\boldsymbol{\theta}_1, \boldsymbol{\theta}_2, \ldots\} \subseteq \boldsymbol{\Theta}$, $\exists \boldsymbol{\theta}'$ such that if $\boldsymbol{\theta}' = \lim_{i \to \infty} \boldsymbol{\theta}_i$ then $\boldsymbol{\theta}' \in \boldsymbol{\Theta}$.*

- *The weight space $\boldsymbol{\Theta}$ is convex such that $\forall \boldsymbol{\theta}_1, \boldsymbol{\theta}_2 \in \boldsymbol{\Theta}$ and $\forall \lambda \in [0, 1]$, we have that $\lambda \boldsymbol{\theta}_1 + (1 - \lambda)\boldsymbol{\theta}_2 \in \boldsymbol{\Theta}$.*

- *The environment observations $\mathbf{x}_t$ are bounded such that $\|\mathbf{x}_t\|_2 < \infty$, $\forall t \in \{0, 1, \ldots, T\}$.*

*Then the loss functions $\mathcal{L}_t^{\mathrm{id}}$ and $\mathcal{L}_t^{\mathrm{ood}}$, dependent on whether $\mathbf{x}_t$ is ID or OOD (as instantiated in Section 3.1), is $\mathbb{K}^{\mathrm{id}}$-Lipschitz and $\mathbb{K}^{\mathrm{ood}}$-Lipschitz for any timestep $t \in \{0, 1, \ldots, T\}$ with*

$$\mathbb{K}^{\mathrm{id}} \leq \big( \sup_{\mathbf{x} \sim \mathcal{P}^{\mathrm{in}}} \|\mathbf{x}\|_2 \big) \cdot \sqrt{\frac{k-1}{k}} \tag{11}$$

$$\mathbb{K}^{\mathrm{ood}} \leq \big( \sup_{\mathbf{x} \sim \mathcal{P}^{\mathrm{out}}} \|\mathbf{x}\|_2 \big) \cdot \sqrt{\frac{\beta^2 (k-1)}{k}} \tag{12}$$

*where $\mathcal{P}^{\mathrm{in}}$ is the environment ID, $\mathcal{P}^{\mathrm{out}}$ is the environment OOD, $\beta$ is a constant hyperparameter, $k$ is the number of ID classes, and $T$ is the total timestep of the online interaction.*

**Proof**   *Given the ID and OOD dependent loss functions as instantiated in Section 3.1, we can reformulate the losses into a single class conditional loss as defined in Equation 13*

$$\mathcal{L}_t(\boldsymbol{\theta}_t) = \mathbb{I}_{\{y_t \neq -1\}} \left( -\log \left( \frac{\exp(f_t^{(y_t)}(\mathbf{x}_t; \boldsymbol{\theta}_t))}{\sum_{j=1}^k \exp(f_t^{(j)}(\mathbf{x}_t; \boldsymbol{\theta}_t))} \right) \right)$$

$$+ \mathbb{I}_{\{y_t = -1\}} \left( -\beta \sum_{i=1}^k \frac{1}{k} \log \left( \frac{\exp(f_t^{(i)}(\mathbf{x}_t; \boldsymbol{\theta}_t))}{\sum_{j=1}^k \exp(f_t^{(j)}(\mathbf{x}_t; \boldsymbol{\theta}_t))} \right) \right) \tag{13}$$

*where $y_t \in \{-1, \ldots, k-1\}$ with $-1$ indicating that the environment sample $\mathbf{x}_t$ is OOD.*

*Consider the case that $\mathbb{I}_{\{y_t \neq -1\}} = 1$ and $\mathbb{I}_{\{y_t = -1\}} = 0$, then our loss function reduces to*

$$\mathcal{L}_t(\boldsymbol{\theta}_t) = -\log \left( \frac{\exp(f_t^{(y_t)}(\mathbf{x}_t; \boldsymbol{\theta}_t))}{\sum_{j=1}^k \exp(f_t^{(j)}(\mathbf{x}_t; \boldsymbol{\theta}_t))} \right) \tag{14}$$

$$= \mathcal{L}_t^{\mathrm{id}}(\boldsymbol{\theta}_t) \tag{15}$$

*Now given the closed and bounded assumptions, we can trivially see that by the mean-value theorem $|\mathcal{L}_t^{\mathrm{id}}(\boldsymbol{\theta}_1) - \mathcal{L}_t^{\mathrm{id}}(\boldsymbol{\theta}_2)| \leq (\sup_{\boldsymbol{\theta} \in \boldsymbol{\Theta}} \|\nabla \mathcal{L}_t^{\mathrm{id}}(\boldsymbol{\theta})\|_F) \cdot \|\boldsymbol{\theta}_1 - \boldsymbol{\theta}_2\|_F$ for any $\boldsymbol{\theta}_1, \boldsymbol{\theta}_2 \in \boldsymbol{\Theta}$. This also implies that $\mathcal{L}_t^{\mathrm{id}}$ is $\mathbb{K}$-Lipschitz continuous with $\mathbb{K} = \sup_{\boldsymbol{\theta} \in \boldsymbol{\Theta}} \|\nabla \mathcal{L}_t^{\mathrm{id}}(\boldsymbol{\theta})\|_F$ as long as $\mathbb{K}$ is bounded.*

*To find the bounds of $\mathbb{K}$, consider the following simple derivation of the derivative with respect to the model output logit $f_t^{(l)}$*

$$\frac{\partial}{\partial f_t^{(l)}} \mathcal{L}_t^{\text{id}}(\boldsymbol{\theta}_t) = -\frac{\partial}{\partial f_t^{(l)}} \log \left( \frac{\exp(f_t^{(y_t)}(\mathbf{x}_t; \boldsymbol{\theta}_t))}{\sum_{j=1}^k \exp(f_t^{(j)}(\mathbf{x}_t; \boldsymbol{\theta}_t))} \right) \tag{16}$$

$$= -\frac{1}{\sigma_t^{(y_t)}} \cdot \frac{\partial}{\partial f_t^{(l)}} (\sigma_t^{(y_t)}) \text{ where } \sigma_t^{(i)} = \frac{\exp(f_t^{(i)}(\mathbf{x}_t; \boldsymbol{\theta}_t))}{\sum_{j=1}^k \exp(f_t^{(j)}(\mathbf{x}_t; \boldsymbol{\theta}_t))} \tag{17}$$

$$= -\frac{1}{\sigma_t^{(y_t)}} \cdot \sigma_t^{(y_t)} (\mathbb{1}_{l=y_t} - \sigma_t^{(l)}) = \sigma_t^{(l)} - \mathbb{1}_{l=y_t} \tag{18}$$

*The derivative of $\mathcal{L}_t^{\text{id}}$ with respect to model parameters $\boldsymbol{\theta}_t$ can be written as*

$$\frac{\partial}{\partial \boldsymbol{\theta}_t} \mathcal{L}_t^{\text{id}}(\boldsymbol{\theta}_t) = \mathbf{x}_t \cdot \frac{\partial \mathcal{L}_t^{\text{id}}}{\partial f} = [\mathbf{x}_t^{(1)}, \dots, \mathbf{x}_t^{(d)}]^\top [\sigma_t^{(0)} - \mathbb{1}_{0=y_t}, \dots, \sigma_t^{(k)} - \mathbb{1}_{k=y_t}] \tag{19}$$

*Furthermore we can also see that the supremum of the ID loss gradient $(\sup_{\boldsymbol{\theta} \in \Theta} \|\nabla \mathcal{L}_t^{\text{id}}(\boldsymbol{\theta})\|_F)$ is achieved only when the softmax of the logits is uniform $\sigma_t = [\frac{1}{k}, \dots, \frac{1}{k}]$. Observing the supremum*

$$\sup_{\boldsymbol{\theta} \in \Theta} \|\nabla \mathcal{L}_t^{\text{id}}(\boldsymbol{\theta})\|_F = \sup_{\boldsymbol{\theta} \in \Theta} \left( \sqrt{\sum_{i=1}^d \sum_{j=1}^k (\mathbf{x}_t^{(i)})^2 (\sigma_t^{(j)} - \mathbb{1}_{j=y_t})^2} \right) \tag{20}$$

$$= \sqrt{\sum_{i=1}^d (\mathbf{x}_t^{(i)})^2} \cdot \sup_{\boldsymbol{\theta} \in \Theta} \left( \sqrt{\sum_{j=1}^k (\sigma_t^{(j)} - \mathbb{1}_{j=y_t})^2} \right) \tag{21}$$

$$= \|\mathbf{x}_t\|_2 \cdot \sqrt{\frac{k-1}{k^2} + (\frac{1}{k} - 1)^2} \tag{22}$$

$$\leq (\sup_{\mathbf{x} \sim \mathcal{P}^{\text{in}}} \|\mathbf{x}\|_2) \sqrt{\frac{k-1}{k}} \quad \forall t \in \{0, 1, \dots, T\} \tag{23}$$

*Consider the case that $\mathbb{I}_{\{y_t \neq -1\}} = 0$ and $\mathbb{I}_{\{y_t = -1\}} = 1$, then our loss function reduces to*

$$\mathcal{L}_t(\boldsymbol{\theta}_t) = -\beta \sum_{i=1}^k \frac{1}{k} \log \left( \frac{\exp(f_t^{(i)}(\mathbf{x}_t; \boldsymbol{\theta}_t))}{\sum_{j=1}^k \exp(f_t^{(j)}(\mathbf{x}_t; \boldsymbol{\theta}_t))} \right) \tag{24}$$

$$= \mathcal{L}_t^{\text{ood}}(\boldsymbol{\theta}_t) \tag{25}$$

*Observing the following derivation of the derivative with respect to the model logit output $f_t^{(l)}$*

$$\frac{\partial}{\partial f_t^{(l)}} \mathcal{L}_t^{\text{ood}}(\boldsymbol{\theta}_t) = -\beta \cdot \frac{\partial}{\partial f_t^{(l)}} \sum_{i=1}^k \frac{1}{k} \cdot \log \left( \frac{\exp(f_t^{(i)}(\mathbf{x}_t; \boldsymbol{\theta}_t))}{\sum_{j=1}^k \exp(f_t^{(j)}(\mathbf{x}_t; \boldsymbol{\theta}_t))} \right) \tag{26}$$

$$= -\frac{\beta}{k} \cdot \frac{\partial}{\partial f_t^{(l)}} \left( \sum_{i=1}^k f_t^{(i)}(\mathbf{x}_t; \boldsymbol{\theta}_t) - \log \left( \sum_{j=1}^k \exp(f_t^{(j)}(\mathbf{x}_t; \boldsymbol{\theta}_t)) \right) \right) \tag{27}$$

$$= -\frac{\beta}{k} \left( 1 - k \cdot \frac{\partial}{\partial f_t^{(l)}} \log \left( \sum_{j=1}^k \exp(f_t^{(j)}(\mathbf{x}_t; \boldsymbol{\theta}_t)) \right) \right) \tag{28}$$

$$= -\frac{\beta}{k} \left( 1 - k \cdot \left( \frac{\exp(f_t^{(l)}(\mathbf{x}_t; \boldsymbol{\theta}_t))}{\sum_{j=1}^k \exp(f_t^{(j)}(\mathbf{x}_t; \boldsymbol{\theta}_t))} \right) \right) \tag{29}$$

$$= -\frac{\beta}{k} \left( 1 - k \cdot \sigma_t^{(l)} \right) \text{ where } \sigma_t^{(l)} = \frac{\exp(f_t^{(l)}(\mathbf{x}_t; \boldsymbol{\theta}_t))}{\sum_{j=1}^k \exp(f_t^{(j)}(\mathbf{x}_t; \boldsymbol{\theta}_t))} \tag{30}$$

*with the derivative of the OOD loss with respect to model parameters $\boldsymbol{\theta}_t$ as*

$$\frac{\partial}{\partial \boldsymbol{\theta}_t} \mathcal{L}_t^{\text{ood}}(\boldsymbol{\theta}_t) = \mathbf{x}_t \cdot \frac{\partial \mathcal{L}_t^{\text{ood}}}{\partial f} = [\mathbf{x}_t^{(1)}, \dots, \mathbf{x}_t^{(d)}]^\top [\frac{\beta}{k}(k \cdot \sigma_t^{(0)} - 1), \dots, \frac{\beta}{k}(k \cdot \sigma_t^{(k)} - 1)] \quad (31)$$

*We can also see that the supremum of the OOD loss gradient ($\sup_{\boldsymbol{\theta} \in \boldsymbol{\Theta}} \|\nabla \mathcal{L}_t^{\text{ood}}(\boldsymbol{\theta})\|_F$) is achieved only when $\sigma_t$ is a one-hot vector (for example $\sigma_t = [1, 0, \dots, 0]$). Therefore the supremum of the OOD loss gradient can be found as*

$$\sup_{\boldsymbol{\theta} \in \boldsymbol{\Theta}} \|\nabla \mathcal{L}_t^{\text{ood}}(\boldsymbol{\theta})\|_F = \sup_{\boldsymbol{\theta} \in \boldsymbol{\Theta}} \left( \sqrt{\sum_{i=1}^{d} \sum_{j=1}^{k} (\mathbf{x}_t^{(i)})^2 \left( \beta \sigma_t^{(j)} - \frac{\beta}{k} \right)^2} \right) \quad (32)$$

$$= \|\mathbf{x}_t\|_2 \cdot \sup_{\boldsymbol{\theta} \in \boldsymbol{\Theta}} \left( \sqrt{\sum_{j=1}^{k} \left( \beta \sigma_t^{(j)} - \frac{\beta}{k} \right)^2} \right) \quad (33)$$

$$= \|\mathbf{x}_t\|_2 \cdot \sqrt{\frac{\beta^2(k-1)}{k^2} + \left( \beta - \frac{\beta}{k} \right)^2} \quad (34)$$

$$= \|\mathbf{x}_t\|_2 \cdot \sqrt{\frac{\beta^2(k-1)}{k}} \quad (35)$$

$$\leq \left( \sup_{\mathbf{x} \sim \mathcal{P}^{\text{out}}} \|\mathbf{x}\|_2 \right) \cdot \sqrt{\frac{\beta^2(k-1)}{k}} \quad \forall t \in \{0, 1, \dots, T\} \quad (36)$$

*Finally, given that at every timestep $\mathcal{L}_t(\boldsymbol{\theta}_t)$ must be strictly equal to $\mathcal{L}_t^{\text{id}}(\boldsymbol{\theta}_t)$ or $\mathcal{L}_t^{\text{ood}}(\boldsymbol{\theta}_t)$, we know that $\mathcal{L}_t^{\text{id}}(\boldsymbol{\theta}_t)$ must be $\mathbb{K}^{\text{id}}$-Lipschitz continuous and $\mathcal{L}_t^{\text{ood}}(\boldsymbol{\theta}_t)$ must be $\mathbb{K}^{\text{ood}}$-Lipschitz continuous $\forall t \in \{0, 1, \dots, T\}$ with*

$$\mathbb{K}^{\text{id}} \leq \left( \sup_{\mathbf{x} \sim \mathcal{P}^{\text{in}}} \|\mathbf{x}\|_2 \right) \cdot \sqrt{\frac{k-1}{k}} \quad (37)$$

$$\mathbb{K}^{\text{ood}} \leq \left( \sup_{\mathbf{x} \sim \mathcal{P}^{\text{out}}} \|\mathbf{x}\|_2 \right) \cdot \sqrt{\frac{\beta^2(k-1)}{k}} \quad (38)$$

**Theorem B.2 (Regret Bound)** *The regret for* SODA *can be upper bounded by*

$$regret = \sum_{t=1}^{T} \mathcal{L}_t(\boldsymbol{\theta}_t) - \min_{\boldsymbol{\theta} \in \boldsymbol{\Theta}} \sum_{t=1}^{T} \mathcal{L}_t(\boldsymbol{\theta}) \quad (39)$$

$$\leq \sqrt{TB} \cdot \left( (\sup_{\mathbf{x} \sim \mathcal{P}^{\text{in}}} \|\mathbf{x}\|_2) \sqrt{\frac{(1 - \tilde{\pi})(k-1)}{k}} + (\sup_{\mathbf{x} \sim \mathcal{P}^{\text{out}}} \|\mathbf{x}\|_2) \sqrt{\frac{\tilde{\pi} \beta^2(k-1)}{k}} \right) \quad (40)$$

$$\approx o(T) \quad (41)$$

*where $\tilde{\pi} = (\sum_{t=1}^{T} \mathbb{I}_{\{y_t=-1\}})/T$ is an empirical estimate of the mixture ratio, $\beta$ is a constant hyperparameter, $k$ is the number of ID classes, $T$ is the total timestep of the online interaction, and $\mathcal{L}_t(\boldsymbol{\theta}_t)$ is defined by equation 13.*

**Proof** *Consider a sequence of observations $\{\tilde{\mathbf{x}}_1, \tilde{\mathbf{x}}_2, \dots, \tilde{\mathbf{x}}_T\}$ generated by a hostile adversarial environment such that regret is maximized.*

*Therefore, given the fixed adversarial samples $\{\tilde{\mathbf{x}}_1, \tilde{\mathbf{x}}_2, \dots, \tilde{\mathbf{x}}_T\}$, we can denote the sequence of loss functions suffered by the model as $\{\tilde{\mathcal{L}}_1, \tilde{\mathcal{L}}_2, \dots, \tilde{\mathcal{L}}_T\}$ where*

$$\tilde{\mathcal{L}}_t = \mathcal{L}_t(\boldsymbol{\theta}_t; \tilde{\mathbf{x}}_t) \quad (42)$$

*Furthermore, as $\mathcal{L}_t(\boldsymbol{\theta})$ is convex $\forall \boldsymbol{\theta} \in \boldsymbol{\Theta}$, the adversarial loss function $\tilde{\mathcal{L}}_t(\boldsymbol{\theta})$ must be similarly convex. Therefore, it must be such that*

$$\tilde{\mathcal{L}}_t(\boldsymbol{\theta}^*) \geq \lim_{\epsilon \to 0} \left( \frac{1}{\epsilon} \tilde{\mathcal{L}}_t(\boldsymbol{\theta}_t + \epsilon(\boldsymbol{\theta}^* - \boldsymbol{\theta}_t)) - \frac{1-\epsilon}{\epsilon} \tilde{\mathcal{L}}_t(\boldsymbol{\theta}_t) \right) \text{ given by convexity} \quad (43)$$

$$\geq \tilde{\mathcal{L}}_t(\boldsymbol{\theta}_t) + \mathbf{tr}(\nabla \tilde{\mathcal{L}}_t(\boldsymbol{\theta}_t) \cdot (\boldsymbol{\theta}^* - \boldsymbol{\theta}_t)) \quad (44)$$

where $\boldsymbol{\theta}^* = \operatorname{argmin}_{\boldsymbol{\theta} \in \boldsymbol{\Theta}} \sum_{t=1}^{T} \tilde{\mathcal{L}}_t(\boldsymbol{\theta})$ and $\mathbf{tr}(\cdot)$ is the matrix trace.

Now, considering $\sum_{t=1}^{T} \|\boldsymbol{\theta}_{t+1} - \boldsymbol{\theta}^*\|_F^2 - \|\boldsymbol{\theta}_t - \boldsymbol{\theta}^*\|_F^2$ we can observe that

$$\sum_{t=1}^{T} \|\boldsymbol{\theta}_{t+1} - \boldsymbol{\theta}^*\|_F^2 - \|\boldsymbol{\theta}_t - \boldsymbol{\theta}^*\|_F^2 \tag{45}$$

$$= \sum_{t=1}^{T} \|\boldsymbol{\theta}_t - \eta \nabla \tilde{\mathcal{L}}_t(\boldsymbol{\theta}_t) - \boldsymbol{\theta}^*\|_F^2 - \|\boldsymbol{\theta}_t - \boldsymbol{\theta}^*\|_F^2 \tag{46}$$

$$= \sum_{t=1}^{T} \left( \sum_{i=1}^{d} \sum_{i=1}^{k} |\boldsymbol{\theta}_t^{(i,j)} - \eta \nabla \tilde{\mathcal{L}}_t(\boldsymbol{\theta}_t)^{(i,j)} - \boldsymbol{\theta}^{*(i,j)}|^2 \right) - \|\boldsymbol{\theta}_t - \boldsymbol{\theta}^*\|_F^2 \tag{47}$$

$$= \sum_{t=1}^{T} \left( \sum_{i=1}^{d} \sum_{i=1}^{k} |\boldsymbol{\theta}_t^{(i,j)} - \boldsymbol{\theta}^{*(i,j)}|^2 + |\eta \nabla \tilde{\mathcal{L}}_t(\boldsymbol{\theta}_t)^{(i,j)}|^2 - 2\eta \nabla \tilde{\mathcal{L}}_t(\boldsymbol{\theta}_t)^{(i,j)} \cdot (\boldsymbol{\theta}_t - \boldsymbol{\theta}^*)^{(i,j)} \right)$$
$$\tag{48}$$
$$\qquad - \|\boldsymbol{\theta}_t - \boldsymbol{\theta}^*\|_F^2$$

$$= \sum_{t=1}^{T} \|\boldsymbol{\theta}_t - \boldsymbol{\theta}^*\|_F^2 + \eta^2 \|\nabla \tilde{\mathcal{L}}_t(\boldsymbol{\theta}_t)\|_F^2 - 2\eta \left( \sum_{i=1}^{d} \sum_{i=1}^{k} \nabla \tilde{\mathcal{L}}_t(\boldsymbol{\theta}_t)^{(i,j)} \cdot (\boldsymbol{\theta}_t - \boldsymbol{\theta}^*)^{(i,j)} \right) \tag{49}$$
$$\qquad - \|\boldsymbol{\theta}_t - \boldsymbol{\theta}^*\|_F^2$$

$$= \sum_{t=1}^{T} \eta^2 \|\nabla \tilde{\mathcal{L}}_t(\boldsymbol{\theta}_t)\|_F^2 - 2\eta \, \mathbf{tr}(\nabla \tilde{\mathcal{L}}_t(\boldsymbol{\theta}_t) \cdot (\boldsymbol{\theta}_t - \boldsymbol{\theta}^*)) \tag{50}$$

Furthermore, we can also observe that

$$\sum_{t=1}^{T} \|\boldsymbol{\theta}_{t+1} - \boldsymbol{\theta}^*\|_F^2 - \|\boldsymbol{\theta}_t - \boldsymbol{\theta}^*\|_F^2 = \|\boldsymbol{\theta}_T - \boldsymbol{\theta}^*\|_F^2 - \|\boldsymbol{\theta}_1 - \boldsymbol{\theta}^*\|_F^2 \tag{51}$$

Therefore, combining equations (50) and (51), we get that

$$\|\boldsymbol{\theta}_T - \boldsymbol{\theta}^*\|_F^2 - \|\boldsymbol{\theta}_1 - \boldsymbol{\theta}^*\|_F^2 \le \sum_{t=1}^{T} \eta^2 \|\nabla \tilde{\mathcal{L}}_t(\boldsymbol{\theta}_t)\|_F^2 - 2\eta \, \mathbf{tr}(\nabla \tilde{\mathcal{L}}_t(\boldsymbol{\theta}_t) \cdot (\boldsymbol{\theta}_t - \boldsymbol{\theta}^*)) \tag{52}$$

$$\le \sum_{t=1}^{T} \eta^2 \|\nabla \tilde{\mathcal{L}}_t^{\mathrm{id}}(\boldsymbol{\theta}_t) \mathbb{I}_{\{y_t \ne -1\}} + \nabla \tilde{\mathcal{L}}_t^{\mathrm{ood}}(\boldsymbol{\theta}_t) \mathbb{I}_{\{y_t = -1\}}\|_F^2$$
$$\qquad - 2\eta \, \mathbf{tr}(\nabla \tilde{\mathcal{L}}_t(\boldsymbol{\theta}_t) \cdot (\boldsymbol{\theta}_t - \boldsymbol{\theta}^*)) \tag{53}$$

$$\le T(1 - \tilde{\pi})(\eta^2 (\mathbb{K}^{\mathrm{id}})^2) + T\tilde{\pi}(\eta^2 (\mathbb{K}^{\mathrm{ood}})^2)$$
$$\qquad - 2\eta \sum_{t=1}^{T} \mathbf{tr}(\nabla \tilde{\mathcal{L}}_t(\boldsymbol{\theta}_t) \cdot (\boldsymbol{\theta}_t - \boldsymbol{\theta}^*)) \tag{54}$$

where $\tilde{\pi} = (\sum_{t=1}^{T} \mathbb{I}_{\{y_t = -1\}})/T$ and, $\forall t \in \{0, 1, \dots, T\}$, $\mathbb{K}^{\mathrm{id}} \ge \|\nabla \tilde{\mathcal{L}}_t^{\mathrm{id}}(\boldsymbol{\theta}_t)\|_F$ and $\mathbb{K}^{\mathrm{ood}} \ge \|\nabla \tilde{\mathcal{L}}_t^{\mathrm{ood}}(\boldsymbol{\theta}_t)\|_F$. Now by reformatting equation 54, we can obtain the following

$$\sum_{t=1}^{T} \mathbf{tr}(\nabla \tilde{\mathcal{L}}_t(\boldsymbol{\theta}_t) \cdot (\boldsymbol{\theta}_t - \boldsymbol{\theta}^*)) \le \frac{\|\boldsymbol{\theta}_1 - \boldsymbol{\theta}^*\|_F^2}{2\eta} - \frac{\|\boldsymbol{\theta}_T - \boldsymbol{\theta}^*\|_F^2}{2\eta} \tag{55}$$

$$+ \frac{T(1 - \tilde{\pi})(\eta (\mathbb{K}^{\mathrm{id}})^2)}{2} + \frac{T\tilde{\pi}(\eta (\mathbb{K}^{\mathrm{ood}})^2)}{2} \tag{56}$$

*Now, observing our regret bound*

$$regret = \sum_{t=1}^{T} \mathcal{L}_t(\boldsymbol{\theta}_t) - \min_{\boldsymbol{\theta} \in \Theta} \sum_{t=1}^{T} \mathcal{L}_t(\boldsymbol{\theta}) \tag{57}$$

$$\leq \sum_{t=1}^{T} \tilde{\mathcal{L}}_t(\boldsymbol{\theta}_t) - \min_{\boldsymbol{\theta} \in \Theta} \sum_{t=1}^{T} \tilde{\mathcal{L}}_t(\boldsymbol{\theta}) \text{ by definition (42)} \tag{58}$$

$$\leq \sum_{t=1}^{T} \tilde{\mathcal{L}}_t(\boldsymbol{\theta}_t) - \tilde{\mathcal{L}}_t(\boldsymbol{\theta}^*) \text{ where } \boldsymbol{\theta}^* = \operatorname*{argmin}_{\boldsymbol{\theta} \in \Theta} \sum_{t=1}^{T} \tilde{\mathcal{L}}_t(\boldsymbol{\theta}) \tag{59}$$

$$\leq \sum_{t=1}^{T} \mathbf{tr}(\nabla \tilde{\mathcal{L}}_t(\boldsymbol{\theta}_t) \cdot (\boldsymbol{\theta}_t - \boldsymbol{\theta}^*)) \text{ by equation (44)} \tag{60}$$

$$\leq \frac{\|\boldsymbol{\theta}_1 - \boldsymbol{\theta}^*\|_F^2}{2\eta} - \frac{\|\boldsymbol{\theta}_T - \boldsymbol{\theta}^*\|_F^2}{2\eta} + \frac{T(1-\tilde{\pi})(\eta(\mathbb{K}^{\mathrm{id}})^2)}{2} + \frac{T\tilde{\pi}(\eta(\mathbb{K}^{\mathrm{ood}})^2)}{2} \text{ by equation (56)} \tag{61}$$

$$\leq \frac{B}{2\eta} + \frac{\eta(T(1-\tilde{\pi})(\mathbb{K}^{\mathrm{id}})^2 + T\tilde{\pi}(\mathbb{K}^{\mathrm{ood}})^2)}{2} \text{ by assumption and } \frac{\|\boldsymbol{\theta}_T - \boldsymbol{\theta}^*\|_2^2}{2\eta} \geq 0 \tag{62}$$

*Thus, consider $\eta = \sqrt{\frac{B}{T(1-\tilde{\pi})(\mathbb{K}^{\mathrm{id}})^2 + T\tilde{\pi}(\mathbb{K}^{\mathrm{ood}})^2}}$, we can see that*

$$regret = \sum_{t=1}^{T} \mathcal{L}_t(\boldsymbol{\theta}_t) - \min_{\boldsymbol{\theta} \in \Theta} \sum_{t=1}^{T} \mathcal{L}_t(\boldsymbol{\theta}) \tag{63}$$

$$\leq \frac{B}{2\eta} + \frac{\eta(T(1-\tilde{\pi})(\mathbb{K}^{\mathrm{id}})^2 + T\tilde{\pi}(\mathbb{K}^{\mathrm{ood}})^2)}{2} = \frac{B + \eta^2 T((1-\tilde{\pi})(\mathbb{K}^{\mathrm{id}})^2 + \tilde{\pi}(\mathbb{K}^{\mathrm{ood}})^2)}{2\eta} \tag{64}$$

$$\leq \sqrt{TB\left( \left( \sup_{\mathbf{x} \sim \mathcal{P}^{\mathrm{in}}} \|\mathbf{x}\|_2^2 \right) \left( \frac{(1-\tilde{\pi})(k-1)}{k} \right) + \left( \sup_{\mathbf{x} \sim \mathcal{P}^{\mathrm{out}}} \|\mathbf{x}\|_2^2 \right) \left( \frac{\tilde{\pi}\beta^2(k-1)}{k} \right) \right)} \text{ by Theorem B.1} \tag{65}$$

$$\leq \sqrt{TB} \cdot \left( (\sup_{\mathbf{x} \sim \mathcal{P}^{\mathrm{in}}} \|\mathbf{x}\|_2) \sqrt{\frac{(1-\tilde{\pi})(k-1)}{k}} + (\sup_{\mathbf{x} \sim \mathcal{P}^{\mathrm{out}}} \|\mathbf{x}\|_2) \sqrt{\frac{\tilde{\pi}\beta^2(k-1)}{k}} \right) \tag{66}$$

$$\approx o(T) \tag{67}$$

## C EXPERIMENTAL DETAILS

In Appendix C.1, we present a detailed description of the OOD datasets that are referenced in Section 4.1. We also include a description of software and hardware specifications in Appendix C.2, training details in Appendix C.3, and evaluation metrics in Appendix C.4.

### C.1 DATASETS

We present a detailed description of the OOD datasets used to evaluate OOD performance and for simulating the online environment.

**OOD Datasets for CIFAR-10:**

- **SVHN** (Netzer et al., 2011a) is a collection of real-world (32×32) images containing cropped house number plates obtained through Google street view images.
- **LSUN Resized** (Yu et al., 2015) is a collection of 10,000 testing images, sampled from the LSUN dataset, spanning across 10 different scenes with images down-sampled to the size of (32×32).
- **Textures** (Cimpoi et al., 2014), or Describable Textures Dataset, is a collection of 5,640 real-world texture images under 47 categories.

- **Places365** (Zhou et al., 2017) contains large-scale photographs of scenes with 365 scene categories. There are 900 images per category in the test set and we leverage a subset of 10,000 images for our experiments.

**OOD Datasets for ImageNet**  We adopt the same OOD test datasets as defined by Huang & Li:

- **iNaturalist** (Van Horn et al., 2018) is a collection of 859,000 plant and animal images spanning over 5,000 different species. Images are selected from 110 classes that are semantically disjoint from ImageNet-1k classes.

- **SUN** (Xiao et al., 2010) is a collection of over 130,000 images of scenes spanning 397 categories. Images are selected from 50 classes that are semantically disjoint from ImageNet-1k classes.

- **Textures** (Cimpoi et al., 2014), or Describable Textures Dataset, is a collection of 5,640 real-world texture images under 47 categories.

- **Places** (Zhou et al., 2017) is a collection of scene images with similar semantic coverage as SUN. Images are selected from a subset of 50 classes that are disjoint from ImageNet-1k classes.

## C.2 SOFTWARE AND HARDWARE

**Software**  We conducted all experiments with Python 3.8.12 and PyTorch 1.10.2.

**Hardware**  All experiments were conducted using NVIDIA GeForce RTX 2080Ti and NVIDIA RTX A6000 graphics processing units.

## C.3 TRAINING DETAILS

During the online interaction, we fixed our learning rate to be $0.0005$ with a batch size of 32, weight decay of $0.0005$, and a default $\beta = 1.0$. We provide an ablation analysis on $\beta$ in Appendix E. We perform full fine-tuning on a ResNet-34 and ResNet-101 model in the CIFAR-10 and ImageNet-1k ID settings respectively. Models are initialized with the models pre-trained on the ID training data split. For pre-training, under the CIFAR-10 ID setting, we leverage the standard softmax cross entropy loss for 200 epochs with a batch size of 64 and a cosine annealing learning rate starting from $0.1$. Under the ImageNet-1k ID setting, we pre-train for 100 epochs with a batch size of 128. For baseline comparison with SSD+ and KNN+, we pre-train with the SupCon (Khosla et al., 2020) loss for 500 epochs, using a batch size of 1024 and a starting learning rate of $0.5$.

In both the stationary and non-stationary settings, we leverage $60\%$ of the OOD dataset for the online mixture environment, and the remaining $40\%$ for test-time evaluation. Note that due to the small dataset size of textures, we allow the online environment to cycle multiple times through textures if necessary. To empirically visualize and measure the regret, we assume that the best model in hindsight is an oracle with no loss at every timestep $t$. This is commonly used for empirically evaluating regret in online learning literature (Alquier et al., 2017; Kivinen et al., 2004; Shui et al., 2023).

## C.4 EVALUATION METRICS

To evaluate our algorithm, we leverage standard metrics in OOD detection literature which includes: (1) the false positive rate of declaring OOD examples as ID when the true positive rate of ID is set as 95% (FPR95 $\downarrow$), and (2) the area under the receiver operating characteristic curve (AUROC $\uparrow$). These measurements are computed based on both the *average* across $T$ rounds of online interaction and the *final* model's performance. Furthermore, we also measure the regret metric in the form of a regret curve over time.

## D  EXTENSION TO THE UNSUPERVISED ONLINE SETTING

In this appendix section, we discuss how to tackle the case where the online environment is unable to provide feedback (i.e. no ground truth labels). In particular, we show how a simple reformulation of SODA's loss function and optimization algorithm can enable SODA to perform *unsupervised online*

| $P_{in}$ (Model) | Method | Evaluated OOD Datasets | | | | | | | | | | Average ID Acc. |
|---|---|---|---|---|---|---|---|---|---|---|---|---|
| | | SVHN | | LSUN-R | | Textures | | Places365 | | Average | | |
| | | FPR ↓ | AUROC ↑ | FPR ↓ | AUROC ↑ | FPR ↓ | AUROC ↑ | FPR ↓ | AUROC ↑ | FPR ↓ | AUROC ↑ | |
| **CIFAR-10** (ResNet-34) | | **Online OOD Detection** | | | | | | | | | | |
| | SODA (Avg) | $2.29^{\pm0.2}$ | $\mathbf{99.65^{\pm0.0}}$ | $1.73^{\pm0.1}$ | $99.74^{\pm0.0}$ | $4.97^{\pm0.1}$ | $\mathbf{99.06^{\pm0.0}}$ | $8.54^{\pm0.9}$ | $97.89^{\pm0.7}$ | $4.38^{\pm0.3}$ | $\mathbf{99.09^{\pm0.2}}$ | $93.49^{\pm0.1}$ |
| | SODA (Final) | $\mathbf{0.15^{\pm0.3}}$ | $\mathbf{99.97^{\pm0.1}}$ | $\mathbf{0.09^{\pm0.1}}$ | $\mathbf{99.97^{\pm0.0}}$ | $\mathbf{2.09^{\pm0.2}}$ | $\mathbf{99.63^{\pm0.1}}$ | $\mathbf{5.33^{\pm1.1}}$ | $\mathbf{98.25^{\pm0.7}}$ | $\mathbf{1.92^{\pm0.4}}$ | $\mathbf{99.46^{\pm0.2}}$ | $93.55^{\pm0.3}$ |
| | | **Online (Unsupervised) OOD Detection** | | | | | | | | | | |
| | USODA (Avg) | $2.38^{\pm0.2}$ | $99.62^{\pm0.0}$ | $1.79^{\pm0.1}$ | $99.71^{\pm0.0}$ | $14.66^{\pm0.4}$ | $97.29^{\pm0.2}$ | $24.86^{\pm0.7}$ | $94.74^{\pm0.4}$ | $10.92^{\pm0.4}$ | $97.84^{\pm0.2}$ | $92.69^{\pm0.1}$ |
| | USODA (Final) | $0.70^{\pm0.1}$ | $99.91^{\pm0.0}$ | $0.15^{\pm0.1}$ | $99.96^{\pm0.0}$ | $10.28^{\pm0.5}$ | $98.06^{\pm0.2}$ | $23.34^{\pm0.8}$ | $94.93^{\pm0.4}$ | $8.62^{\pm0.4}$ | $98.22^{\pm0.2}$ | $92.77^{\pm0.1}$ |
| | | iNaturalist | | SUN | | Textures | | Places | | Average | | |
| **ImageNet-1k** (ResNet-101) | | **Online OOD Detection** | | | | | | | | | | |
| | SODA (Avg) | $2.99^{\pm0.1}$ | $99.27^{\pm0.0}$ | $5.93^{\pm0.2}$ | $98.43^{\pm0.0}$ | $7.76^{\pm0.2}$ | $97.93^{\pm0.0}$ | $9.15^{\pm0.3}$ | $97.60^{\pm0.1}$ | $6.46^{\pm0.2}$ | $98.31^{\pm0.0}$ | $77.85^{\pm0.2}$ |
| | SODA (Final) | $\mathbf{0.71^{\pm0.2}}$ | $\mathbf{99.83^{\pm0.0}}$ | $\mathbf{2.19^{\pm0.3}}$ | $\mathbf{99.45^{\pm0.1}}$ | $\mathbf{2.07^{\pm0.2}}$ | $\mathbf{99.50^{\pm0.0}}$ | $\mathbf{3.12^{\pm0.3}}$ | $\mathbf{99.25^{\pm0.1}}$ | $\mathbf{2.02^{\pm0.3}}$ | $\mathbf{99.51^{\pm0.1}}$ | $77.96^{\pm0.5}$ |
| | | **Online (Unsupervised) OOD Detection** | | | | | | | | | | |
| | USODA (Avg) | $21.73^{\pm0.5}$ | $95.64^{\pm0.3}$ | $25.71^{\pm0.5}$ | $94.60^{\pm0.2}$ | $35.09^{\pm1.3}$ | $91.43^{\pm0.8}$ | $38.27^{\pm0.9}$ | $90.68^{\pm0.7}$ | $30.20^{\pm0.8}$ | $93.09^{\pm0.5}$ | $77.12^{\pm0.2}$ |
| | USODA (Final) | $11.21^{\pm0.7}$ | $97.59^{\pm0.4}$ | $13.87^{\pm0.6}$ | $96.91^{\pm0.3}$ | $24.17^{\pm1.6}$ | $94.81^{\pm1.0}$ | $28.15^{\pm1.1}$ | $92.95^{\pm0.8}$ | $19.35^{\pm1.0}$ | $95.57^{\pm0.6}$ | $77.38^{\pm0.3}$ |

Table 3: **USODA results.** Comparison of OOD detection performance between SODA and USODA under CIFAR-10 and ImageNet-1k benchmarks. All results are averaged over 5 random runs. We report both the average and final performance as specified in Appendix C.4. Superior results are in **bold**.

*OOD detection.* More specifically, we first reformulate the ID loss function in Appendix D.1 and then discuss how we optimize the unsupervised losses in Appendix D.2. Lastly, we present our empirical observations in Appendix D.3.

## D.1 UNSUPERVISED LOSS FUNCTION

**Out-of-distribution Loss Function** We leverage the same loss function $\mathcal{L}_t^{\text{ood}}$ which was previously defined in Equation 2 as our OOD loss. Critically, it is important to note that the OOD loss $\mathcal{L}_t^{\text{ood}}$ is an unsupervised loss function, as it only depends on the model prediction without needing labeling information. We also note that the $\mathcal{L}_t^{\text{ood}}$ can be interpreted as entropy maximization—where maximizing the entropy of the softmax output is the same as minimizing the cross-entropy between the softmax output and the uniform distribution.

**In-distribution Loss Function** We now define the in-distribution loss function as entropy minimization. More specifically, let $\mathcal{L}_t^H$ be our new unsupervised ID loss:

$$\mathcal{L}_t^{\text{H}}(\boldsymbol{\theta}_t) = -\sum_{i=1}^{k} \sigma_t^{(i)}(\mathbf{x}_t; \boldsymbol{\theta}_t) \log(\sigma_t^{(i)}(\mathbf{x}_t; \boldsymbol{\theta}_t)) \text{ where } \sigma_i(\mathbf{x}_t; \boldsymbol{\theta}_t) = \frac{\exp(f_t^{(i)}(\mathbf{x}_t; \boldsymbol{\theta}_t))}{\sum_{j=1}^{k} \exp(f_t^{(j)}(\mathbf{x}_t; \boldsymbol{\theta}_t))} \tag{68}$$

where $f_t^{(i)}(\mathbf{x}_t; \boldsymbol{\theta}_t)$ denoting the $i$-th element of $f_t(\mathbf{x}_t; \boldsymbol{\theta}_t)$. We note that $\mathcal{L}_t^H$ is now an unsupervised loss function where, in combination with $\mathcal{L}_t^{\text{ood}}$, we are min-maxing entropy with the goal of making the model confidence more separable between ID and OOD samples.

## D.2 OPTIMIZING THE UNSUPERVISED LOSS: USODA

To optimize our loss function in the online setting, we continue to leverage online gradient descent. However, instead of choosing to incur ID or OOD loss based on the environment-revealed label, we instead leverage the existing OOD detector and incur loss based on whether the detector believes a sample is ID or OOD. More specifically, we restrict the algorithm to only optimize on a subset of observations that our detector has very confidently deemed as being ID or OOD. Furthermore, to ensure stability throughout online learning, we choose to optimize only a subset of features $\boldsymbol{\theta}'$ which includes the classification head (the penultimate linear layer of the neural network) and any batch normalization layers present in the neural network. We provide a more detailed rundown of the unsupervised extension of SODA (USODA) in Algorithm 3.

---

**Algorithm 3** Unsupervised Stream Out-of-Distribution Adaptation (USODA)

---

1: **Input hyperparameter:** $\eta, \delta_1, \delta_2$
2: Train the initial classifier $f_1 : \mathbb{R}^d \to \mathbb{R}^k$ on the ID task
3: Define the *msp* OOD scoring function $S : \mathbb{R}^k \to [0, 1]$
4: **for** $t = 1, \ldots, T$ **do**
5:     Receive environment instance $\mathbf{x}_t \in \mathbb{R}^d$
6:     Estimate the OOD scores $\hat{s}_t = S(f_t(\mathbf{x}_t))$
7:     **if** $\hat{s}_t \leq \delta_1$ **then**
8:         Suffer OOD loss $\mathcal{L}_t^{\text{ood}}(\mathbf{x}_t)$ defined in Equation 2
9:         Update model parameters $\boldsymbol{\theta}'_{t+1} = \boldsymbol{\theta}'_t - \eta \nabla_{\theta'} \mathcal{L}_t^{\text{ood}}(\mathbf{x}_t)$
10:     **else if** $\hat{s}_t > \delta_2$ **then**
11:         Suffer ID entropy minimization loss $\mathcal{L}_t^{\text{H}}(\mathbf{x}_t)$ defined in Equation 68
12:         Update model parameters $\boldsymbol{\theta}'_{t+1} = \boldsymbol{\theta}'_t - \eta \nabla_{\theta'} \mathcal{L}_t^{\text{H}}(\mathbf{x}_t)$
13:     **end if**
14: **end for**
15: **Return** $f_T$

---

For all of our experiments, the hyperparameters are chosen from $\delta_1 \in \{0.10, 0.15, \ldots, 0.35\}$ and $\delta_2 \in \{0.70, 0.75, \ldots, 0.95\}$.

### D.3 EMPIRICAL EVALUATIONS

We present evaluations in Table 3, comparing SODA and USODA under the stationary CIFAR-10 and ImageNet-1k ID settings. As expected the unsupervised USODA extension is unable to outperform the feedback available SODA method. Nonetheless, we note that USODA still outperforms the best offline OOD detection method found in Table 1. USODA reduces the final average FPR95 by **12.83**% when compared to ReAct in the ImageNet-1k ID setting. We can also see that USODA is comparably better than WOODS. This improved performance further demonstrates the strength and flexibility of our online method for real-world scenarios.

## E FURTHER STUDIES

**Performance on Unseen OOD** In Table 4, we provide detailed OOD detection performances of SODA's final OOD detector on unseen OOD datasets. The OOD detector was adapted with respect to a particular environment ID ($\mathcal{P}^{\text{in}}$) and environment OOD ($\mathcal{P}^{\text{out}}$) dataset. From Table 4, we can see that the OOD detector adapted to one specific environment can still generalize to unseen OOD data. Additionally, we can observe that, there can be significant differences in the final SODA detector's performance on unseen OOD, depending on the specific environment OOD ($\mathcal{P}^{\text{out}}$). In particular, we note that there is a reduction of FPR95 by **25.04**% when Places365 is used as the environment OOD, in contrast to $\mathcal{P}^{\text{out}} = $ SVHN. We hypothesize that this is due to Places365 being a more informative

| $\mathcal{P}^{\text{in}}$ (Model) | Environment OOD $\mathcal{P}^{\text{out}}$ | Unseen OOD Datasets | | | | | | | | | | Average ID Acc. |
|---|---|---|---|---|---|---|---|---|---|---|---|---|
| | | SVHN | | LSUN-R | | Textures | | Places365 | | Average | | |
| | | FPR $\downarrow$ | AUROC $\uparrow$ | FPR $\downarrow$ | AUROC $\uparrow$ | FPR $\downarrow$ | AUROC $\uparrow$ | FPR $\downarrow$ | AUROC $\uparrow$ | FPR $\downarrow$ | AUROC $\uparrow$ | |
| | | Online OOD Detection (SODA) | | | | | | | | | | |
| **CIFAR-10** (ResNet-34) | SVHN | — | — | $35.28^{\pm6.5}$ | $94.53^{\pm1.0}$ | $34.39^{\pm4.6}$ | $94.49^{\pm0.8}$ | $52.52^{\pm4.1}$ | $89.86^{\pm0.9}$ | $40.73^{\pm5.1}$ | $92.96^{\pm0.9}$ | $93.63^{\pm0.3}$ |
| | LSUN-R | $22.99^{\pm6.1}$ | $96.60^{\pm1.0}$ | — | — | $22.29^{\pm2.8}$ | $96.32^{\pm0.5}$ | $43.37^{\pm4.3}$ | $91.64^{\pm0.9}$ | $29.55^{\pm4.4}$ | $94.85^{\pm0.8}$ | $93.74^{\pm0.2}$ |
| | Textures | $12.28^{\pm5.3}$ | $98.01^{\pm0.8}$ | $9.37^{\pm2.7}$ | $98.37^{\pm0.3}$ | — | — | $39.82^{\pm4.4}$ | $91.84^{\pm0.8}$ | $20.49^{\pm4.1}$ | $96.07^{\pm0.5}$ | $93.16^{\pm0.5}$ |
| | Places365 | $30.98^{\pm9.5}$ | $95.12^{\pm1.3}$ | $0.52^{\pm0.1}$ | $99.79^{\pm0.0}$ | $15.58^{\pm1.7}$ | $97.06^{\pm0.3}$ | — | — | $15.69^{\pm3.8}$ | $97.32^{\pm0.4}$ | $93.40^{\pm0.2}$ |
| | | iNaturalist | | SUN | | Textures | | Places | | Average | | |
| | | Online OOD Detection (SODA) | | | | | | | | | | |
| **ImageNet-1k** (ResNet-101) | iNaturalist | — | — | $45.68^{\pm5.7}$ | $87.47^{\pm0.9}$ | $57.49^{\pm4.8}$ | $82.90^{\pm0.9}$ | $51.92^{\pm4.0}$ | $85.88^{\pm0.8}$ | $51.69^{\pm4.8}$ | $85.42^{\pm0.9}$ | $78.72^{\pm0.4}$ |
| | SUN | $26.78^{\pm4.2}$ | $94.11^{\pm0.8}$ | — | — | $60.10^{\pm5.1}$ | $81.26^{\pm0.9}$ | $13.85^{\pm3.7}$ | $96.19^{\pm0.7}$ | $33.58^{\pm4.3}$ | $90.52^{\pm0.8}$ | $78.53^{\pm0.3}$ |
| | Textures | $28.33^{\pm3.8}$ | $93.20^{\pm0.7}$ | $53.18^{\pm5.9}$ | $87.04^{\pm1.0}$ | — | — | $57.47^{\pm4.2}$ | $84.85^{\pm0.8}$ | $46.33^{\pm4.6}$ | $88.36^{\pm0.8}$ | $78.08^{\pm0.6}$ |
| | Places | $12.92^{\pm4.6}$ | $97.26^{\pm0.8}$ | $9.02^{\pm3.5}$ | $97.76^{\pm0.6}$ | $50.84^{\pm4.3}$ | $84.75^{\pm0.8}$ | — | — | $24.26^{\pm4.1}$ | $93.26^{\pm0.7}$ | $78.56^{\pm0.5}$ |

Table 4: **Unseen OOD evaluation.** Evaluation of SODA on unseen OOD datasets where $\mathcal{P}^{\text{test}} \neq \mathcal{P}^{\text{out}}$. We report the final detector performances as specified in Appendix C.4. $\uparrow$ indicates larger values are better, while $\downarrow$ indicates smaller values are better.

and challenging OOD dataset, which may enable the final SODA detector to generalize better to unseen OOD datasets.

**Effect of $\beta$ Hyperparameter** In Figure 2, we present an ablation on the effect of the $\beta$ hyperparameter, which modulates the weight of the OOD loss function. Specifically, a higher $\beta$ indicates a larger penalty while a smaller $\beta$ indicates a smaller penalty on the OOD loss. For simplicity, we consider the stationary environment where $\pi = 0.2$ and results are averaged over all OOD test datasets in the CIFAR-10 ID setting. From Figure 2, we can observe that our online OOD detection performance remains consistent for $\beta \in (0, 1]$. In the case where $\beta = 0$, we can observe a degradation in OOD detection performance. This is expected given that $\beta = 0$ implies we are not learning from the OOD observations.

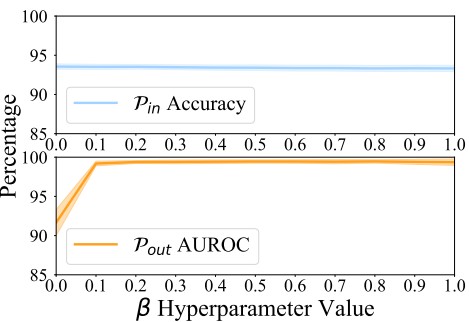

Figure 2: **Effect of $\beta$ hyperparameter** on ID classification (blue) and OOD detection (orange).

**Effect of Mixture Ratio** In Figure 3, we present an ablation on the effect of the mixture ratio $\pi_t$. Recall in Section 2, $\pi_t$ modulates the fraction of OOD data in the environment distribution. Note that a smaller $\pi_t$ reflects the more realistic scenario where the majority of online data should remain ID. For simplicity, we consider the stationary environment where $\pi_t$ does not change with time (effectively reduces to $\pi$) and results are averaged over all OOD test datasets in the CIFAR-10 ID setting. From Figure 3, we observe that SODA's performance remains stable and strong with increasing $\pi$. As expected, in the extreme case of very large $\pi$, a significant degradation happens for ID classification accuracy. We hypothesize that this is likely due to over-adaptation to the OOD data (which

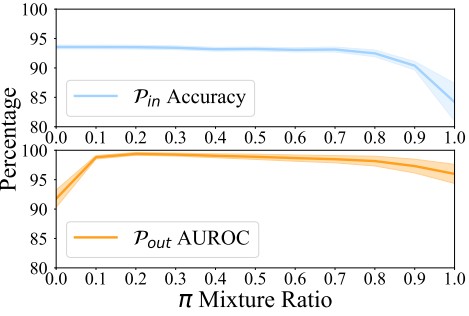

Figure 3: **Effect of mixture ratio $\pi$** on ID classification (blue) and OOD detection (orange).

dominates in the online environment), and as a result, underfits the ID data that infrequently occurs.

**Effect of $\Delta$ Hyperparameter** In Figure 4, we study the effect of $\Delta$ on the regret. Recall in Section 4.1, we defined $\Delta$ as the switch time in the non-stationary setting where the environment shifts after every $\Delta$ timesteps to a new environment OOD dataset. We can observe, from Figure 4, that a smaller $\Delta$ leads to a larger overall regret. This is expected as the smaller $\Delta$ indicates a more difficult problem—the environment is constantly shifting making adaption to the environment OOD more challenging. Despite the difficult setting ($\Delta = 800$), we can see that the performance of SODA is still trending sub-linearly. This sub-linear trend, in the challenging smaller $\Delta$ settings, is encouraging as it indicates that SODA is not limited by very frequent fluctuations in the online environment OOD.

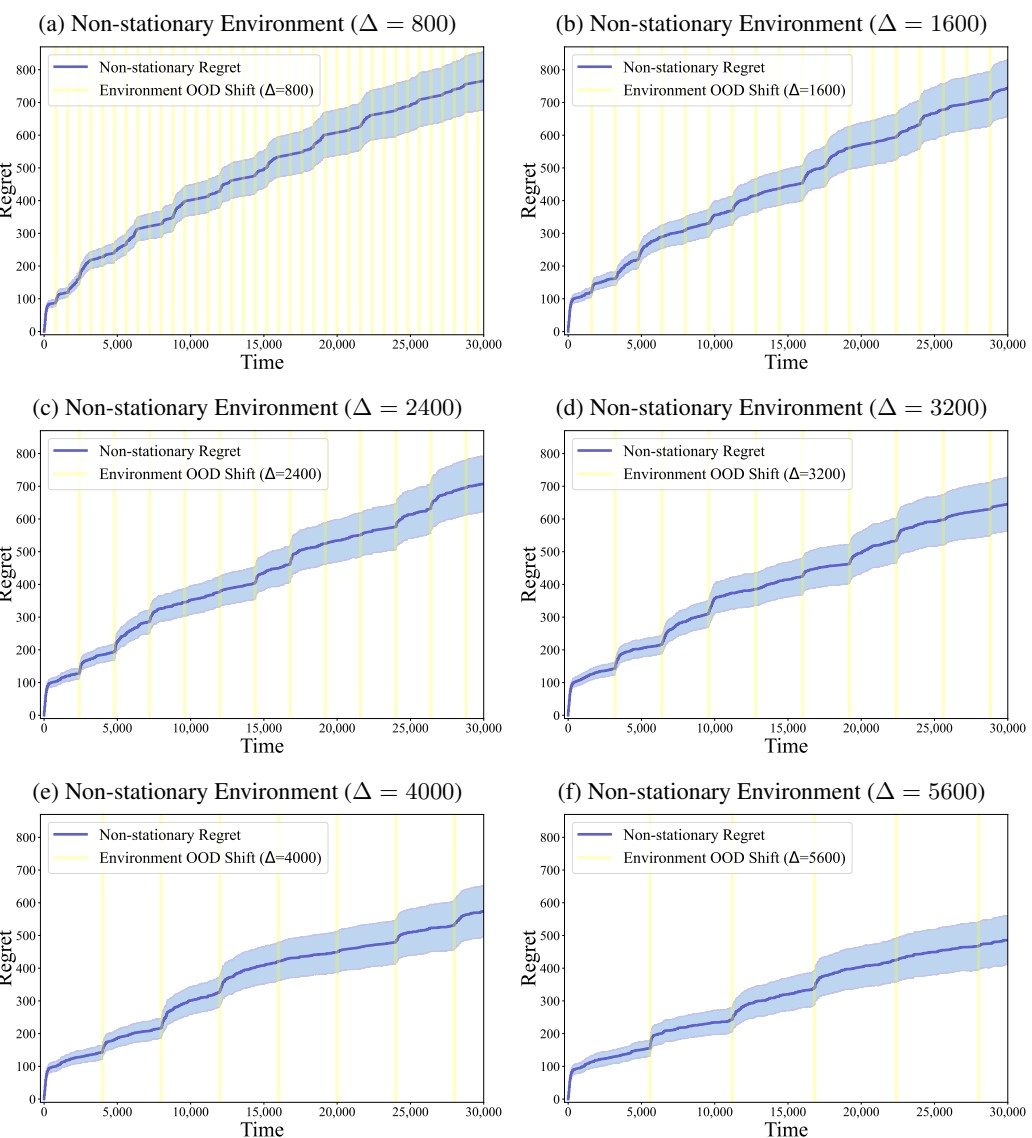

Figure 4: **Effect of $\Delta$** on regret (blue). We denote every timestamp with an OOD distribution change through a (yellow) vertical line.

