# OpenReview forum: "SODA: Stream Out-of-Distribution Adaptation"
_ICLR.cc/2024/Conference — Submitted to ICLR 2024_

### Official Review · Reviewer_oqWZ · 2023-10-29

**Soundness:** 3 good
**Presentation:** 3 good
**Contribution:** 2 fair
**Rating:** 3
**Confidence:** 4

**Summary:**

This paper considers the problem of online learning with out-of-distribution data. In this setting, the learning model is required to learn with streaming data with OOD data, given whether a data is OOD is known. The proposed method constructs the loss function by combining the ID loss and the OOD loss, and updates the model via online gradient descent. The authors also analyze the static regret of the proposed algorithm. Empirical results show the empirical success of the proposed method.

**Strengths:**

1. This paper is well organized and easy to follow.

2. The proposed algorithm is simple and easy to implement with many OOD loss and shows empirical success on several benchmark datasets.

**Weaknesses:**

1. The proposed algorithm requires strong supervision feedback on whether each instance is OOD data in the online learning procedure, which is difficult to satisfy in real-world applications. In this problem setting (whether an instance is OOD data is available), the learning task is then over-simplified. Such a strong supervision assumption makes the proposed method straightforward, which is a simple extension of online learning algorithms with a specific loss function, that is, combine of ID loss and OOD loss in this draft.
While the authors also offer an unsupervised version that does not require environmental feedback and shows promising empirical results, it lacks solid theoretical guarantees, which limits the applicability of the proposed method.

2. The theoretical results in this work are well established in the online learning literature, which limits the technical contribution of this research. These results are relatively straightforward to obtain when a specific convex loss function is specified. In this draft, it is the combination of the ID loss and the OOD loss. It's also worth noting that the literature typically uses adaptive regret or dynamic regret to capture the non-stationarity of the environment, as opposed to the static regret used in this draft.


3. Regarding the implementation of the algorithm, determining the hyperparameter $\lambda$ for OOD detection before and during the online learning process seems to be challenging. Furthermore, the authors do not elaborate on how $\lambda$ is determined in the experiments or how it affects the algorithm's performance. There is also no published code.

**Questions:**

How is $\lambda$ in the algorithm determined in the experiments and how does $\lambda$ affect the performance of the algorithm?

---

> ### Author Response · Authors · 2023-11-18
> **Response to reviewer oqWZ**
>
> We want to express our gratitude to the reviewer for their pertinent comments. In the sections below, we have provided some additional clarifications regarding the specific questions raised by the reviewer.
>
> **Concerns regarding Supervised Feedback**
>
> > The reviewer's concern regarding the supervised feedback requirement of SODA is understandable, and it is a large part of why we designed the no-feedback unsupervised extension to SODA. However, we would like to direct the reviewer to consider several other factors regarding the supervised online learning setting. Although obtaining feedback might appear challenging at first glance, there are many real-world scenarios where relevant feedback is provided during online interaction. For instance, in cases where the model aims to predict whether an image posted online receives impressions, clicks, or content violation reports (OOD), the environment will eventually provide labels based on future data, indicating whether the image indeed garners impressions/clicks or triggers content violation (OOD) reports. Moreover, there may also be cases where implementor-provided feedback is given during the online interaction. Finally, even in the worst-case scenario where no feedback is possible, the unsupervised extension (USODA) can still be leveraged, which demonstrates very strong performance on the same comprehensive set of evaluations as SODA.
>
> **Selecting for Lambda Thresholding Hyperparameter**
>
> > We understand the reviewer's confusion, as comprehending where the lambda threshold lies in our evaluations can be perplexing. Throughout our work, we’ve consistently adhered to established norms in OOD detection. In our online simulation, we internally utilize the widely adopted OOD score, when the true positive rate of a validation ID set is 95%, as the lambda threshold. However, it is crucial to note that this threshold is neither utilized nor required when evaluating SODA/USODA or any prior OOD detection works. In OOD detection literature, the selection of the lambda threshold is generally arbitrary and left to the implementer's discretion. For a more comprehensive look at this, we encourage the reviewer to refer to Appendix C.4. Additionally, further information on standard thresholding practices can also be found in [1, 2]. Should any lingering confusion persist, we would also be happy to provide further clarification.
>
> **Significance and motivation of theory**
>
> > Our primary objective with this work is to address a significant gap in modern Out-of-Distribution (OOD) detection, where OOD detection is often confined to the static supervised setting. Therefore, the goal of the theoretical analysis is to showcase how to leverage the classic online learning analysis for the specific use case of online OOD detection. Furthermore, in online OOD, our model is learning to be uncertainty-aware, and built to function as a detector for semantically shifted OOD samples. In other words, standard online learning analysis differs from our work in that, in standard online learning, no OOD detection occurs throughout the online interaction, and the model is not learning to be more uncertainty-aware. Thus, we believe our theoretical contribution is valuable and complementary to existing online learning literature.
>
> [1] Dan Hendrycks and Kevin Gimpel. A baseline for detecting misclassified and out-of-distribution examples in neural networks. In 5th International Conference on Learning Representations, ICLR 2017, 2017.
>
> [2] Jingkang Yang, Kaiyang Zhou, Yixuan Li, and Ziwei Liu. Generalized out-of-distribution detection: A survey. arXiv preprint arXiv:2110.11334, 2021.

---

### Official Review · Reviewer_dt7Y · 2023-11-01

**Soundness:** 3 good
**Presentation:** 2 fair
**Contribution:** 2 fair
**Rating:** 3
**Confidence:** 3

**Summary:**

This paper studies stream learning with out-of-distribution (OOD) data with $T$ rounds. At each round, In each round, the learner needs to predict whether the incoming unlabeled samples are from the OOD class. After that, the label of the sample is revealed to the learner. The paper proposes an online learning algorithm, which updates the model by performing a gradient step with the cross entropy loss.  Regret analysis and extensive experiments are conducted to validate the proposed methods.

**Strengths:**

The strengths of the paper are as follows:
- Problem setting: The problem of Out-Of-Distribution (OOD) detection with stream data is a relevant and interesting topic for the reliable machine learning community.
- Presentation: The paper is well-structured and clearly written in most parts.
- Experiments: Extensive experiments have been conducted in this paper to validate the proposed methods.

**Weaknesses:**

The weaknesses of this paper are as follows:
- Originality and Significance: My main concern about the paper is the originality of the proposed method. It seems to me that Algorithm 2 (and Algorithm 3 in the Appendix) is a direct application of the classical Online Gradient Descent (OGD) algorithm [1, Chapter 3.1] with the cross-entropy loss. The regret analysis for OGD is somewhat standard. Although the authors have provided some related work discussion, I am unconvinced by the claim "...our framework differs in that we deal with non-stationary OOD data in the context of OOD detection." For the classical online convex optimization framework, the loss function can be arbitrarily different at each round (under certain boundedness assumptions); the framework can already be used for non-stationary data.

- About the Feedback: In the main paper, the algorithm is developed based on the assumption that the labels of the OOD data are available after the learner has made the prediction. This seems like a strong assumption to me since the main challenge of OOD detection lies in the lack of supervision of OOD data. When the labeled OOD data are available, I am confused about what the difference is between the proposed framework and supervised online learning. Although the authors show in the Appendix that one can also use unsupervised loss functions to perform OOD detection, it is unclear how the proposed unsupervised loss is (theoretically) guaranteed to output a well-performing OOD detector, which makes the method less appealing.

- About the Theoretical Analysis: I think the statement of Theorem 3.1 is somewhat informal, as the conditions of the loss functions and parameter settings are not provided. It seems to me that the analysis in the Appendix crucially relies on the convexity of the loss function, and thus, the linear model is applied. However, in the main paper, the algorithm is equipped with a neural network. I think it would be better if the authors could clearly mention the discrepancy between the theory and algorithm implementation.

**Questions:**

- Could you provide more discussion on the differences between the proposed methods (Algorithms 2 and 3) and the online learning algorithm (e.g., OGD)?
- Could we have certain guarantees for the unsupervised OOD detection loss $\mathcal{L}_t^H$? It seems to me one can also apply similar arguments as Theorem 2 to obtain a regret bound in terms of $\mathcal{L}_t^H$, but it is unclear why minimizing $\mathcal{L}_t^H$ can lead to a well-performed detector that can eventually minimize the true loss function $\mathcal{L}_t$.
- As shown by the equation above (63), it seems to me that the setting of the learning rate $\eta$ requires the knowledge of $T$, $\mathbb{K}^{id}$ and $\mathbb{K}^{ood}$, which are unknown in practice. It would be better if the authors could provide more discussions on how to select the parameters for the algorithm.

---

> ### Author Response · Authors · 2023-11-18
> **Response to reviewer dt7Y (1/2)**
>
> We thank the reviewer for their valuable comments and have thoroughly considered all the feedback provided. In the sections below, we have addressed some of the comments raised by the reviewer.
>
> **Originality and differences with Online Learning**
>
> > We would like to offer several new perspectives for the reviewer to consider as they engage with our work. In contrast to traditional OOD detection methods that operate in the static supervised setting without leveraging any online observations, and traditional online learning methods that do not adequately address semantically shifted OOD, our work is designed to bridge this gap and provide insights into effective OOD detection in an online setting [2].
> > We recognize that leveraging online learning to address OOD detection might seem straightforward. However, we want to emphasize that (1) this problem has never been previously addressed and (2) this problem offers a necessary contribution for liberating existing OOD detection away from the confines of its static supervised setting. Additionally, the Online OOD regime is disjoint from both OOD detection and online learning, presenting unique and previously unaddressed challenges such as:
> > - How do we define an online mixture environment embedded with semantically shifted out-of-distribution?
> > - What does "non-stationary" mean in the context of semantically shifted OOD and OOD detection?
> > - What is the most appropriate way to incrementally adjust OOD detectors while preserving model classification in the online setting?
> > - How do the unsupervised and supervised regimes differ, and is it even feasible to build an algorithm with both unsupervised ID and OOD observations?
> > - Lastly, how do we appropriately evaluate Online OOD methods, and what metrics/datasets should we employ?
>
> **Response on Non-stationarity**
>
> > The reviewer raises an excellent point, regarding the fact that traditional online convex optimization (OCO) already addresses non-stationary environments. However, we would like to note that the non-stationary environments defined in traditional OCO significantly differ from the non-stationary setting we address in SODA. Crucially, our online OOD detection framework specifically targets cases where a non-stationary, semantically shifted out-of-distribution is presented alongside an implementation of an OOD detector. More specifically, the goal of traditional OCO is to correctly classify/generalize observations as the non-stationary in-distribution environment undergoes covariate distributional shifts. In contrast, our online OOD detector aims to detect observations, that should not be classified into any in-distribution classes, as the non-stationary out-of-distribution undergoes semantic shifts. In summary, our primary objective in online OOD is to address the uncertainty awareness of non-stationary, semantically shifted OOD observations. While traditional OCO focuses on the classification of non-stationary or covariate-shifted in-distribution observations.
>
> **USODA and Supervised Feedback**
>
> > The reviewer raises an excellent point regarding the strong OOD feedback assumption in SODA. Although obtaining feedback may appear challenging at first glance, there are many real-world scenarios where feedback is available during online interaction. For instance, in cases where the model aims to predict whether an image posted online receives impressions, clicks, or content violation reports (OOD), the environment will eventually provide labels based on future data, indicating whether the image indeed garners impressions/clicks or triggers content violation (OOD) reports. Similarly, any human-in-the-loop environment would also have readily available feedback. Even in the worst-case scenario where no feedback is possible, we provided an unsupervised extension to SODA (USODA). However, given that the unsupervised method lacks clear guarantees on its OOD estimates, it is challenging for us to provide strong theoretical results without making stronger assumptions on either the robustness of the model or the underlying online environment. Nevertheless, USODA still demonstrates exceptional performance on the same comprehensive set of evaluations as SODA (Appendix D).  Additionally, the goal with the L^H_t loss function is to function alongside L^{ood}_t to min-max entropy, thereby making the model confidence more separable between ID and OOD samples. Although it may be initially confusing how minimizing entropy can correlate with minimizing the loss, there are actually many prior test-time adaptation works that already leverage and provide great insights on this question [1].

---

> ### Author Response · Authors · 2023-11-18
> **Response to reviewer dt7Y (2/2)**
>
> **Selecting for Hyperparameters**
>
> > While the selection of hyperparameters may seem challenging at first, in practice, choosing the learning rate is actually quite straightforward. In OOD detection, even without online OOD learning, we are guaranteed to have some degree of OOD detection capabilities. This is especially true if one opts to use the stronger OOD detectors as showcased in Section 4.3. Therefore, in practice, it boils down to choosing a learning rate that is not excessively large relative to the model architecture, as the online OOD detection method will still function even with a suboptimal small learning rate. For example, we've found that as long as the learning rate stays lower than 1e-3, the algorithm will generally perform optimally.
>
> **Linear Assumption in the Theorem**
>
> > The reviewer raises a valid point regarding the assumption of linear classifiers in Theorem 3.1. The theorem has its empirical relevance. Specifically, our guarantee ensures that all linear probing methods, a common approach used throughout online and continual learning, will achieve optimality. Furthermore, it is worth mentioning that linear probing of the final layer is largely employed in our unsupervised extension of SODA. Thus, our findings should still hold practical value in our USODA algorithm and various online applications.
>
> [1] Dequan Wang, Evan Shelhamer, Shaoteng Liu, Bruno Olshausen, and Trevor Darrell. Tent: Fully test-time adaptation by entropy minimization. In 9th International Conference on Learning Representations, ICLR 2021, 2021.
>
> [2] Jingkang Yang, Kaiyang Zhou, Yixuan Li, and Ziwei Liu. Generalized out-of-distribution detection:
> A survey. arXiv preprint arXiv:2110.11334, 2021.

---

### Official Review · Reviewer_eZ4h · 2023-11-01

**Soundness:** 3 good
**Presentation:** 3 good
**Contribution:** 2 fair
**Rating:** 3
**Confidence:** 3

**Summary:**

A novel online OOD detection framework is presented in this work to better adapt to a dynamic environment. This is crucial for its use in real-world applications. From the theoretical aspect, this study demonstrates that the proposed algorithm provably achieves sub-linear regret and converges to the optimal OOD detector over time. The proposed algorithms are also validated by empirical evaluations on commonly used offline OOD data as well as its corresponding online version (simulated).

**Strengths:**

This paper solves a significant and challenging research question, i.e., online OOD detection.
This paper contributes with both theoretical and empirical insights.

**Weaknesses:**

The solution seems less technical to me. This needs further discussion during rebuttal.
The current version seems to embed an OOD detector into a naive online learning framework.

**Questions:**

1. This paper claims its novelty in online OOD detection. However, [1] is also designed for online OOD detection. Please compare the difference, and specify one what aspect this work is better than [1].
[1] Wu, Xinheng, et al. "Meta OOD Learning For Continuously Adaptive OOD Detection." Proceedings of the IEEE/CVF International Conference on Computer Vision. 2023.

2. Could you specify which items in (2) and (3) are designed specially for an online non-stantioanry environment?

3. In the current SODA, OOD detector and classifier are updated at every new t. However, this update is not always good. Please explain why it is necessary to update OOD detector and classifier for EVERY new t, rather than retain the previous one for some t.

4. Will SODA have a high computational cost? Please also specify the computational cost for update process.

---

> ### Author Response · Authors · 2023-11-18
> **Response to reviewer eZ4h (1/2)**
>
> We thank the reviewer for dedicating their time and providing insightful comments. In the sections below, we have added responses to the questions raised by the reviewer.
>
> **Concerns with the Online Framework’s Simplicity**
>
> > The reviewer raises a valid concern and we would like to rephrase some sections of our work that may not have been fully expressed in the initial reading. We want to clarify that our primary objective with SODA is to address a significant gap in modern OOD detection literature, specifically the need for a concrete formalization on how to enable OOD detection in the online setting. Given the limited amount of relevant literature in this area, the base SODA method was designed to serve as a simple yet rigorous realization of our online OOD framework, with the capability of being adaptable to other existing OOD scoring functions (Section 4.3). In short, we presented SODA to provide a necessary initial formalization for the problem introduced in online OOD. Additionally, while the base SODA method offers an algorithmically simple solution to the online OOD problem, we encourage the reviewer to read the unsupervised extension to SODA (USODA) presented in Appendix D. The unsupervised extension places SODA in a more distinct and challenging unsupervised online framework. Sadly, due to space constraints, much of our unsupervised algorithm has been moved to the Appendix. Nevertheless, we hope that the USODA algorithm will better clarify the uniqueness of our problem setting, and showcase how our formalizations can guide new solutions for tackling the online OOD problems.
>
> **Comparison with CAOOD**
>
> > We thank the reviewer for bringing to our attention a recent work on continuously adaptive OOD detection (CAOOD) [1]. After further reading, we would like to highlight several points to help clarify the differences between CAOOD and our work.
> > - A primary goal of our work is to provide a necessary initial framework to better define the problem of online OOD. However, the framework setting as proposed in CAOOD, differs significantly from online OOD detection. In particular, the distributional shifts observed in the CAOOD are limited to small gradual distributional variations, which puts implicit constraints on what observations are in the online environment [1].
> > - We also want to clarify that the initial pre-training setting for MOL is significantly different from our work. In particular, MOL requires a model to be pre-trained through a specific training regime, meaning that any existing models would need to be retrained in order to operate with MOL [1]. In other words, MOL is not pre-training agnostic while both SODA and its unsupervised extension (USODA) are agnostic to how the initial model is pre-trained. Furthermore, MOL also requires virtual OOD synthesis to generate OOD samples during initial pre-training, which further alters the pre-training algorithm and adds additional computational complexity.
> > - Additionally, SODA has the benefit of being incredibly versatile and able to function across various OOD scoring functions (ODIN & energy), while MOL is mostly limited to just the energy scoring function. Our paper also incorporates theoretical analyses on regret that are not present in MOL, helping further distinguish the contributions of our work.
> > - We would also like to emphasize that the experiments in MOL differ from the experiments presented in our work. For example, many of the evaluations are limited to small gradual distributional shifts over time on low-resolution CIFAR datasets [1].
> > - Lastly, we want to clarify that the objective of our work is to provide a framework for future research to address the challenges of online OOD. While there may be similarities in the underlying topic, SODA and MOL consider different frameworks, settings, and methodologies for addressing this issue. Therefore, we believe there should be minimal concern regarding overlap, with each work contributing unique insights to the broader topic of OOD detection.

---

> ### Author Response · Authors · 2023-11-18
> **Response to reviewer eZ4h (2/2)**
>
> **Other Questions**
>
> > The items in Sections 2 and 3 that specifically pertain to the non-stationary setting include the environment setup in Section 2.1 and the environment declaration in Section 3. In particular, both the distribution P^{out}_t and \pi_t enable a non-stationary environment, as the distribution and its mixture ratio can change throughout the online interaction. The reviewer raises a valid question regarding the optimality of mandating an update at every timestep t. Indeed, in cases where feedback is unavailable or noisy, updating at every timestep t may be suboptimal. However, we would like to direct the reviewer to our unsupervised extension (USODA) presented in Appendix D, where this requirement of updating at every timestep is no longer strictly true. But for the supervised SODA algorithm, given that the supervised setting provides ground truth feedback, adding an update to occur every timestep t should not hurt the model's overall performance. Additionally, the update process incurs minimal computational cost as it involves just a gradient step for both the supervised SODA and unsupervised USODA methods.
>
> [1] Xinheng Wu, Jie Lu, Zhen Fang, and Guangquan Zhang. Meta OOD Learning For Continuously Adaptive OOD Detection. In Proceedings of the IEEE/CVF International Conference on Computer Vision, pp. 19353-19364. 2023.

---

> > ### Comment · Reviewer_eZ4h · 2023-11-22
> > **Thanks for the replies to my questions**
> >
> > I would like to thank the authors' replies to my questions.
> >
> > Their answers alleviated my concerns to a very limited extent. It seems that the authors themselves do not have a clear idea of their contribution to OOD or online learning, considering they are taking the contents in the Appendix as the unique advantage of the proposed method. Honestly, Appendix D does not show impressive findings from my perspective. It lacks good insights into the significance of SODA or USODA.
> >
> > > "our primary objective with SODA is to address a significant gap in modern OOD detection literature, specifically the need for a concrete formalization on how to enable OOD detection in the online setting."
> > Given [1] filled the gap of online OOD detection to some extent with formal definitions and loss, I could barely recognize the novelty of "concrete formalization on how to enable OOD detection in the online setting" as is claimed.
> >
> > The authors emphasize their theoretical contributions. However, they even couldn't comprehensively compare this paper with [1] on a theoretical level. Indeed, [1] provides a better profounding mathematical understanding from my perspective, with consistent mathematical formulations.
> >
> > It should be also noted that I would consider this paper as working on the topic of ood detection rather than online learning. How your regret analysis significantly benefits ood detection is unclear. I would consider most learning schema that updates outdated models could fit the online setting you are claiming. If have you provided such a generic setting, you should also provide powerful statistical/mathematical support at the same time.

---

### Official Review · Reviewer_WYay · 2023-11-09

**Soundness:** 3 good
**Presentation:** 3 good
**Contribution:** 2 fair
**Rating:** 5
**Confidence:** 2

**Summary:**

This paper analyzes the out-of-distribution detection problem. It presents an online detection algorithm that keeps updating the model parameters as the stream reveals data.
The algorithm is simple, yet demonstrated experimentally to be effective.

**Strengths:**

The main strengths of this work are:

- The paper is simple and easy to follow and understand. It gently introduces the problem and smoothly translates to the proposed method.

- The proposed algorithm (SODA) is simple and neat.

- The provided experimental results shows the effectiveness of the proposed SODA compared to other baselines under the streaming evaluation.

**Weaknesses:**

The main weaknesses of this work are:

1- Paper writing: While the paper is generally easy to follow in sections (1, 2, 3), there are several missing details (for a non-expert) that made reading the experimental section harder. such as:

(1a) Definition of performance measures such as FPR and AUROC.

(1b) The rationale behind choosing the datasets in Table 1. Why I SCHN considered OOD compared to CIFAR? Is there a way to quantify how far the evaluated distribution is from the training one?

2- The experiments conducted on this work consider stationary datasets such as ImageNet and CIFAR-10 where the stream is synthetically constructed. I would advise tackling realistic benchmarks such as [A, B, C] from the online learning literature where the stream is defined with respect to time.

3- Discussion of limitation: The proposed SODA, while being effective in many cases, can fail in other scenarios. For instance, consider a stream with very small visual variations with respect to time (e.g. a survallance camera). SODA will be then presented with the same batch (with small variations) repeatedly for some time. Would this make the online updates overfit the network parameters and fail to generalize on new novel batches? A discussion on the limitations of SODA should be included.

[A] Online Continual Learning with Natural Distribution Shifts, ICCV 2021.

[B] The CLEAR Benchmark: Continual LEArning on Real-World Imagery, NeruIPS 2021.

[C] Drinking from a Firehose: Continual Learning with Web-scale Natural Language, TPAMI 2023.

**Questions:**

Please refer to the weaknesses section.

---

> ### Author Response · Authors · 2023-11-18
> **Response to reviewer WYay**
>
> We would like to thank the reviewer for their insightful comments. In the sections below, we have addressed in detail all the questions raised by the reviewer. Furthermore, we are also happy to address any other inquiries the reviewer may have.
>
> **Clarifications on Evaluation Metrics and Datasets**
>
> > We understand the reviewer's confusion, and would like to provide some additional background to help clarify the reviewer’s questions regarding evaluation metrics and datasets. Throughout our work, all evaluation metrics, and datasets found in Table 1, closely follow the standard used in OOD detection literature. More specifically, FPR refers to the false positive rate of declaring OOD examples as ID when the true positive rate of ID is set at 95%, and AUROC refers to the area under the receiver operating characteristic curve. A more detailed explanation of these metrics, and the specific OOD datasets employed, can be found in Appendices C.4 and C.1. Figure 1(b) also illustrates an experiment involving an OOD stream defined with respect to time. Works [1, 2] may also be helpful for better contextualizing the metrics and datasets used in OOD detection. Given that our goal with SODA is to provide a means to move OOD detection beyond its traditional offline supervised setting, we felt it is important to align closely with the standards used in OOD detection. This alignment can also help to facilitate a clear and easy path for incorporating the online environment into future OOD detection works.
>
> **Limitations under Domain Adaptation**
>
> > The reviewer raises an excellent question regarding the potential limitation of SODA under small distributional shifts over time. In the traditional OOD detection setting, we define OOD as any observation that undergoes a significant semantic shift from the in-distribution. This means that the small distributional shifts, such as the covariate shifts observed in surveillance footage, may not be considered out-of-distribution in the OOD detection setting. In other words, our goal in OOD detection is to detect semantically shifted observations rather than engaging with covariate distributional shifts. Therefore, the reviewer's question about generalizing to online covariate OOD shifts, would primarily fall within the domain adaptation regime, which is beyond the scope of our current work. Nonetheless, there are many existing works that deal with this problem of life-long domain adaptation [3, 4] and these techniques present existing solutions that can be integrated with our methods to mitigate any anticipated domain adaptation issues.
>
> [1] Dan Hendrycks and Kevin Gimpel. A baseline for detecting misclassified and out-of-distribution
> examples in neural networks. In International Conference on Learning Representations, 2017.
>
> [2] Jingkang Yang, Kaiyang Zhou, Yixuan Li, and Ziwei Liu. Generalized out-of-distribution detection:
> A survey. arXiv preprint arXiv:2110.11334, 2021.
>
> [3] Judy Hoffman, Trevor Darrell, and Kate Saenko. Continuous manifold based adaptation for evolving visual domains. In Proceedings of the IEEE conference on computer vision and pattern recognition, pp. 867-874. 2014.
>
> [4] Riccardo Volpi, Diane Larlus, and Grégory Rogez. Continual adaptation of visual representations via domain randomization and meta-learning. In Proceedings of the IEEE/CVF Conference on Computer Vision and Pattern Recognition, pp. 4443-4453. 2021.

---

### Author Response · Authors · 2023-11-18
**Global Response to Reviewers’ Feedback**

We would like to express our gratitude to all the reviewers for their invaluable and insightful feedback. Notably, it is exciting to hear that our work addresses a crucial problem (*eZ4h*, *dt7Y*) through engaging methodologies and theories (*WYay*, *eZ4h*). Moreover, we are thrilled that reviewers found strong empirical success, demonstrated by our work, across a wide set of experiments (*WYay*, *dt7Y*, *oqWZ*). Additionally, we are also delighted to learn that our work is well-written and easy to follow (*WYay*, *dt7Y*, *oqWZ*).

We have addressed the comments from each reviewer with individualized responses in the sections below. Once again, we thank the reviewers for the time and effort invested in providing detailed and insightful feedback.

---

### Meta-Review · Area_Chair_VZts · 2023-12-06

**Metareview:**

The paper provides a technique to handle out of distribution (OOD) examples in a streaming setting. The reviewers indicate the presentation is good and the paper is easy to follow. The setting is well motivated and the experiments shows a great promise to the proposed technique. The main concerns raised were: (1) novelty: oqWZ, dt7Y eZ4h mention the increment over existing literature to be limited, (2) applicability: dt7Y and oqWZ considered the assumption of having feedback about OOD examples to be too strong, WYay proposed having benchmarks that better reflect realistic streaming settings.
The concerns raised are quite major, and they were disputed by the reviewers during the discussion period. Given this, I do not think the paper in its current form can be published in ICLR

**Justification For Why Not Higher Score:**

the paper lacks sufficient novelty

**Justification For Why Not Lower Score:**

n/a

---

### Decision · Program_Chairs · 2024-01-16

Reject